# Inorganic nanosheets facilitate humoral immunity against medical implant infections by modulating immune co-stimulatory pathways

Chuang Yang [1,4], Yao Luo[1,4], Hao Shen[1,4], Min Ge[2], Jin Tang[3], Qiaojie Wang[1], Han Lin [2] ✉, Jianlin Shi [2] ✉ & Xianlong Zhang [1] ✉

Strategies to manipulate immune cell co-inhibitory or co-activating signals have revolutionized immunotherapy. However, certain immunologically cold diseases, such as bacterial biofilm infections of medical implants are hard to target due to the complexity of the immune co-stimulatory pathways involved. Here we show that two-dimensional manganese chalcogenophosphates $MnPSe_3$ (MPS) nanosheets modified with polyvinylpyrrolidone (PVP) are capable of triggering a strong anti-bacterial biofilm humoral immunity in a mouse model of surgical implant infection via modulating antigen presentation and costimulatory molecule expression in the infectious microenvironment (IME). Mechanistically, the PVP-modified MPS (MPS-PVP) damages the structure of the biofilm which results in antigen exposure by generating reactive oxidative species, while changing the balance of immune-inhibitory (IL4I1 and CD206) and co-activator signals (CD40, CD80 and CD69). This leads to amplified APC priming and antigen presentation, resulting in biofilm-specific humoral immune and memory responses. In our work, we demonstrate that pre-surgical neoadjuvant immunotherapy utilizing MPS-PVP successfully mitigates residual and recurrent infections following removal of the infected implants. This study thus offers an alternative to replace antibiotics against hard-to-treat biofilm infections.

Medical implants which have revolutionized medicine such as orthopedic implants and catheters, are threatened by the risk of microbial biofilm formation at an unprecedented rate. Invading pathogens within biofilm shelters are highly resistant to biocides and antibiotic chemotherapies as well as host immune defenses, resulting in persistent microbial infections with a negative impact on public health and global economies[1]. Biofilm elicits a hypoxic, nutrient-deprived, and acidic microenvironment which polarizes the host immune cells towards anti-inflammatory phenotypes[2]. Recently, the targeted modulation toward host immune defense is becoming the cutting-edge research for anti-biofilm strategies in the post-antibiotic era[3].

[1]Department of Orthopaedics, Shanghai Jiao Tong University Affiliated Sixth People's Hospital, Shanghai Jiao Tong University, Shanghai 200233, P. R. China. [2]State Key Laboratory of High Performance Ceramics and Superfine Microstructures, Shanghai Institute of Ceramics Chinese Academy of Sciences, Research Unit of Nanocatalytic Medicine in Specific Therapy for Serious Disease, Chinese Academy of Medical Sciences, Shanghai 200050, P. R. China. [3]Department of Clinical Laboratory, Shanghai Jiao Tong University Affiliated Sixth People's Hospital, Shanghai 200233, P. R. China. [4]These authors contributed equally: Chuang Yang, Yao Luo, Hao Shen. ✉e-mail: linhan@mail.sic.ac.cn; jlshi@mail.sic.ac.cn; dr_zhangxianlong@sjtu.edu.cn

As for immunomodulatory strategies, a one-side regulation focusing on innate immune cells such as neutrophils and macrophages featured insufficient anti-biofilm effects, which failed to generate effective adaptive anti-biofilm immunity[4]. Therefore, the next generation of infectious immunotherapy may be catalyzed by inducing targeting neutralizing antibodies and long-lasting memory responses against biofilm[5]. But there are several barriers in designing adaptive immunity modulatory strategies for typical biofilm eradication. These include the poorly immunogenic bacterial-associated antigens (BAA) in the biofilm fortress, which would result in deficient antigen presentation and subsequent T cell priming and activation[6,7]. The biofilm formation generates a niche of immune depression by modeling an immunologically cold microenvironment, including exhaustion of antigen-presenting cells (APC), invalidity of T cell priming, insufficient somatic hypermutations in B cells, and waning of antibody responses. Besides, infectious biofilm elicits abnormal activation of immune suppressive cells such as regulatory T cells ($T_{reg}$)[2,8–11]. This misled adaption of immune response is closely linked to the imbalance of co-inhibitory and co-activating signals in the humoral immunity pathways, which finally confined the number of therapeutically active antibodies engaging in biofilms[4,5,12–15].

The last decade has witnessed great progress in nanotechnology for immunotherapy with intrinsic immunomodulatory functions, optimized antigen presentation, persistent immune responses induction, and complex engagement in the immune pathways[16,17]. In particular, inorganic nanoadjuvants have been extensively studied towards next-generation vaccines[18]. Nanomaterials containing manganese (Mn), selenium (Se), or phosphorus (P) are potential immune-stimulatory candidates according to previous studies[19–21]. In addition, we previously devised 2D niobium carbide MXene[22] and silicon nanosheets[23] for the treatment of implant infections. Hence, we suppose the two-dimensional manganese chalcogenophosphates MnPSe$_3$ nanosheet (MPS NS) can act as an inorganic immune-stimulatory agent for the treatment of implant infections.

Here we show that PVP-modified MPS (MPS-PVP) enhances BAAs exposure in biofilm and potentiates antibacterial immune responses of macrophages via inhibiting immune-inhibitory signals and activating co-activator signals. This alternation promotes macrophages to present endogenous bacterial antigens and initiate subsequent anti-bacterial biofilm humoral immunity in a mouse model of implant infections. MPS-PVP also boosts bacteria-specific memory B cells and targeting antibody responses, thereby preventing postsurgical infection relapse. This study thus opens up possibilities for individualized antigen-specific immunotherapy for the management of recalcitrant implant infections in the post-antibiotic era, the schematic illustration of MPS-PVP enabled anti-bacterial biofilm immunotherapy is shown in Fig. 1.

## Results

### Synthesis and characterization of MnPSe$_3$ nanosheets

The typical crystalline structure of MPS NSs is presented in Fig. 2a. MPS NSs were synthesized by sonication-assisted liquid delamination from pristine manganese chalcogenophosphates (or manganese phosphorus triselenide) MnPSe$_3$ crystals. Scanning electron microscopy (SEM) image of bulk MPS exhibits well-stacked, laminated microstructure (Fig. 2b). Transmission electron microscopy (TEM) analysis demonstrates MPS NSs of typical sheet-like morphology (Fig. 2c). The Raman spectra of bulk MPS and MPS nanosheets were shown in Fig. 2d. For bulk MPS, the strongest peak at 218.4 cm$^{-1}$ marked the stretching vibration mode of the PSe$_3$ unit. The another $A_{1g}$ phonon mode at 146.3 cm$^{-1}$ and two additional $E_g$ phonon modes at 154.6 and 171.1 cm$^{-1}$ are originated from the vibration of [P$_2$Se$_6$]$^{4-}$ units. For MPS nanosheets, the phonon vibration modes are downshifted and weakened obviously. These results are in accord with the few-layered MPS NSs in reported literatures[24]. The elemental mappings and EDS profile of the MPS NSs sample demonstrate a uniform distribution of Mn, P and Se components (Fig. 2e), which is consistent with the EDS profile (Supplementary Fig. S1). Atomic force microscopy (AFM) analysis provides further evidence for the morphology of freestanding MPS nanosheets (Fig. 2f). The thickness of MPS NSs measured by AFM exhibits the typical height of ~3 nm (Fig. 2g), while the lateral size is statistically distributed from 120 to 240 nm as determined by TEM image (Fig. 2h). Mn$^{2+}$ exhibited sensitive catalytic activity towards endogenous H$_2$O$_2$ via Fenton-like reaction[25], thus, we examined the Fenton catalytic reaction of MPS in acidic infectious microenvironment (IME) (Fig. 2i). Upon adding MPS into the acid conditions, the absorbance around 422 nm significantly declines with the pHs elevating, presenting that the acidity enhanced Fenton-like process. To guarantee a desired colloidal stability of MPS in physiological medium, the surface modification of polyvinylpyrrolidone (PVP) on MPS samples was performed (MPS-PVP). Further incubation of MC3T3-E1 cells with MPS-PVP indicates negligible oxidative damage of MPS-PVP towards normal cells (Supplementary Fig. S2).

### MPS-PVP modulates biofilm antigen expression

Biofilms contribute to the resistance of bacteria to immune cells, which increases the recalcitrance of implant infections to conventional treatments. Bacterial biofilm evolves multiple mechanisms to evade immune recognition and antigen presentation[2,7], including the formation of persister cells, modulation of antigen expression, and the secretion of protective extracellular polymeric substances (EPS). Therefore, activating the release of biofilm antigens marks a pivotal issue to profit biofilm-targeted immunotherapy. Herein, the potential of MPS-PVP with the concentration of 10 and 20 µg mL$^{-1}$ (denoted as MPS1 and MPS2, respectively) to elicit biofilm antigen expression was compared with the classic antibiotics (Ant, 20 µg mL$^{-1}$) of vancomycin (Van) and Ofloxacin, which are used in clinic as first-line therapies toward infections caused by Staphylococcus aureus (S. aureus) and Escherichia coli (E. coli). MPS-PVP induces severe destruction of biofilm integrity and significant bacterial death, demonstrated by fluorescent staining for biofilm (Fig. 3a). The destruction of biofilm structural integrity and EPS is essential for bacterial antigen release and recognition by immune cells. Quantitative analysis reveals a significant reduction of biofilm biomass for groups of MPS1 and MPS2 (Fig. 3b). SEM observations further confirmed the decline of biofilm density and the structural deformation of bacterial cell wall after MPS-PVP treatments (Fig. 3c).

The reduction in biofilm is in positive correlation with enhanced antigens release which can be better accessed by immune cells. Flow cytometric analysis indicated MPS-PVP induced higher PI positive rate, suggesting enhanced bacterial apoptosis (Fig. 3d, f). The cell debris of apoptotic and damaged bacteria are vital sources of bacterial antigens for cross-presentation which are avidly internalized by dendritic cells (DC)[26]. In addition, a higher level of ROS production in MPS-PVP-treated biofilm group is observed (Fig. 3e, g), which contributes to the possible mechanism of biofilm destruction induced by MPS. According to previous studies, DNA degradation and lipid peroxidation are the immediate cause of bacterial death induced by ROS[27–29]. We thereby hypothesized that the ability of MPS-PVP to kill bacteria was likely attributed to DNA degradation and lipid peroxidation. To test this hypothesis, we determined the degree of bacterial genomic DNA degradation after incubation with MPS-PVP in simulated infectious conditions. A notable DNA cleavage effect was demonstrated after MPS-PVP treatments (Supplementary Fig. S3). In addition, the MDA content of bacteria exposed to various treatments was quantified. The results showed a statistically significant increase in bacterial MDA level in MPS-PVP groups (Supplementary Fig. S4), indicative of a robust lipid peroxidation.

These studies manifest that MPS-PVP could effectively break the biofilm barrier and leverage biofilm as endogenous antigen deposit for the further activation of adaptive anti-biofilm immunity (Fig. 3h).

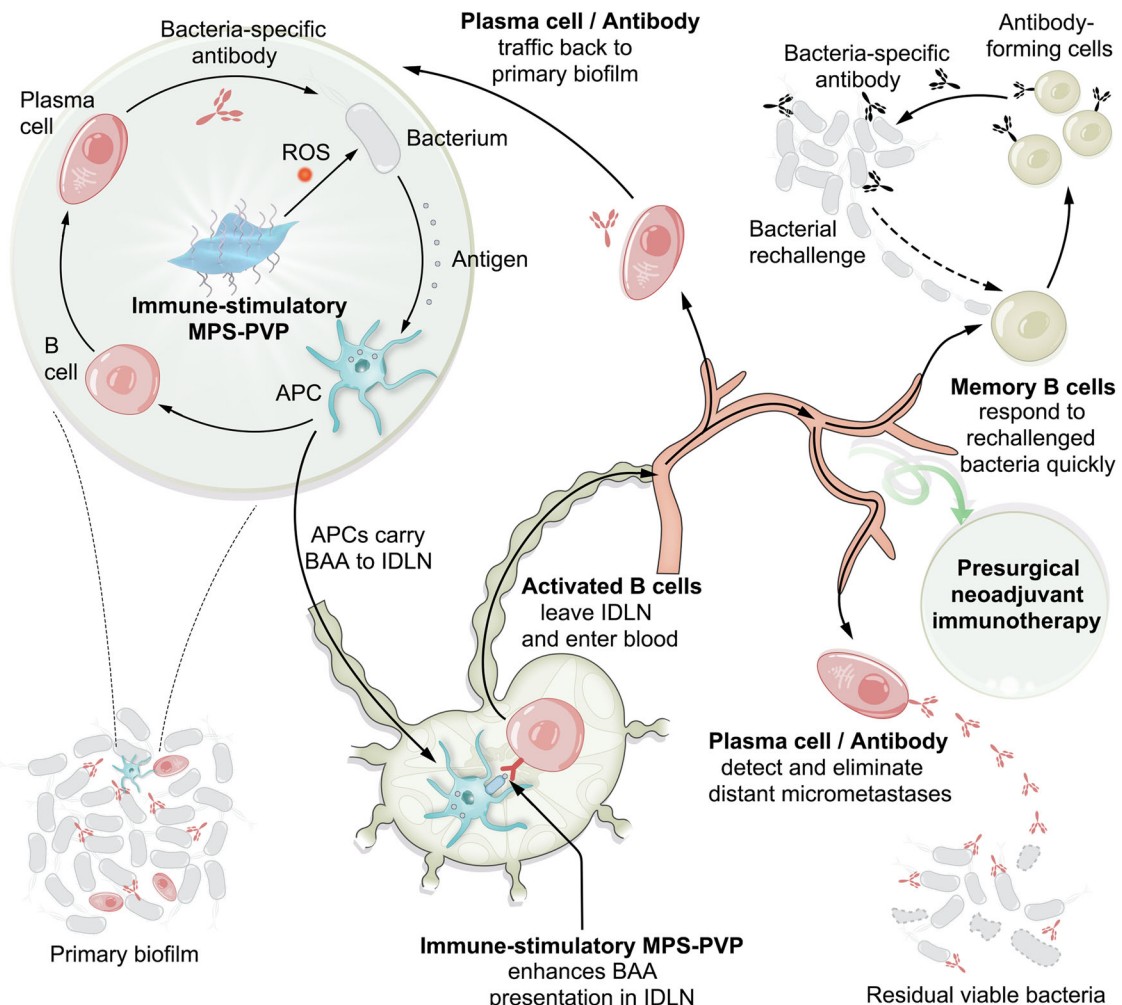

**Fig. 1 | Schematic illustration of immune-stimulatory MPS-PVP for eliciting anti-biofilm humoral immunity and achieving presurgical neoadjuvant immunotherapy.** Immune-stimulatory MPS-PVP leads to "in situ" expansion of bacterial-specific B cells and antibodies in primary biofilm (left). MPS induces bacterial-associated antigens (BAA) expression by producing ROS via Fenton-like reaction. Antigen-presenting cells (APC) pick up and process the BAAs for further presentation. Biofilm-infiltrating B cells would be primed then and matured into plasma cells for bacteria-specific antibody production. Immune-stimulatory MPS-PVP facilitates and amplifies these immune responses. Besides, BAA-containing APCs traffic to infection-draining lymph nodes (IDLN), where they present antigens to bacteria-specific B cells. At this point immune-stimulatory MPS-PVP plays a role of promoting the activation of bacteria-specific B cells. Thus activated B cells mature into either long-lived plasma cells that constantly release antigen-specific antibodies or long-lived memory B cells, and then take part in the blood circulation. Part of plasma cells and antibodies may migrate back to the biofilm, while others would seek out micrometastatic infection sites and clear up the residual bacteria after surgery. In the meantime, memory B cells will persist in the host and quickly transform into antibody-producing cells upon detecting re-challenged bacteria. These immune responses could be applied as pre-surgical neoadjuvant immunotherapy for preventing infection relapse (right).

## MPS-PVP activates co-stimulatory signaling pathways in RAW264.7 macrophages

The induction of immune-inhibitory signals and inhibition of co-activator signals in the pathways of APCs including bacterial recognition, phagocytosis, as well as antigen processing and presentation, represents a vital means of biofilm immune evasion[15,30]. Therefore, we profiled the changes in immune-inhibitory and co-activator signals of RAW264.7 macrophages (a typical APC cell type) treated by MPS-PVP using RNA-seq. The differentially expressed genes (DEG) elicited by MPS-PVP treatment were fistly identified (Fig. 4a). The pathways matching bacterial recognition and antigen presentation were most differentially expressed in cells treated with MPS-PVP (Fig. 4b). It could be found that the NOD-like receptor signaling, TNF signaling and IL-17 signaling pathway were expressed in high activity in MPS-PVP group, indicating effective stimulation of major co-activator signaling pathways (Fig. 4c). Biofilm skews the host immune response with the help of co-inhibitory signals in APCs and lymphocytes[15,31]. Interrogation of bulk RNA-seq profiles based on KEGG database

disclosed that the expression of immune-inhibitory molecules, such as interleukin-4-induced-1 (IL4I1) and CD206, was downregulated in MPS-PVP group. In the contrary, MPS-PVP upregulated the expression of co-activator molecules, such as CD40, CD80 and CD69. Besides, a network of pro-inflammatory genes associated with anti-biofilm, such as TNFα and IL1β, were upregulated in MPS-PVP group (Fig. 4d). IL4I1 and CD206 are immune-inhibitory molecules that modulate the duration and amplitude of T cell response, the activity of which would hamper adaptive immunity[32–35]. On the contrary, CD40, CD80, and CD69 are vital co-activator molecules, which can license APCs to enhance T cell activation and B cell priming[14,36]. Pro-inflammatory cytokines also play crucial roles in activating DCs maturation and priming T cell activation. Gene-gene interactions of cell adhesion molecules highlighted CD40 and CD80 as key immune checkpoints of the immunosuppressive biofilm (Fig. 4e). Together, these transcriptomic profiles evidenced that MPS-PVP can change the balance of immune-inhibitory and co-activating signaling pathways in macrophages.

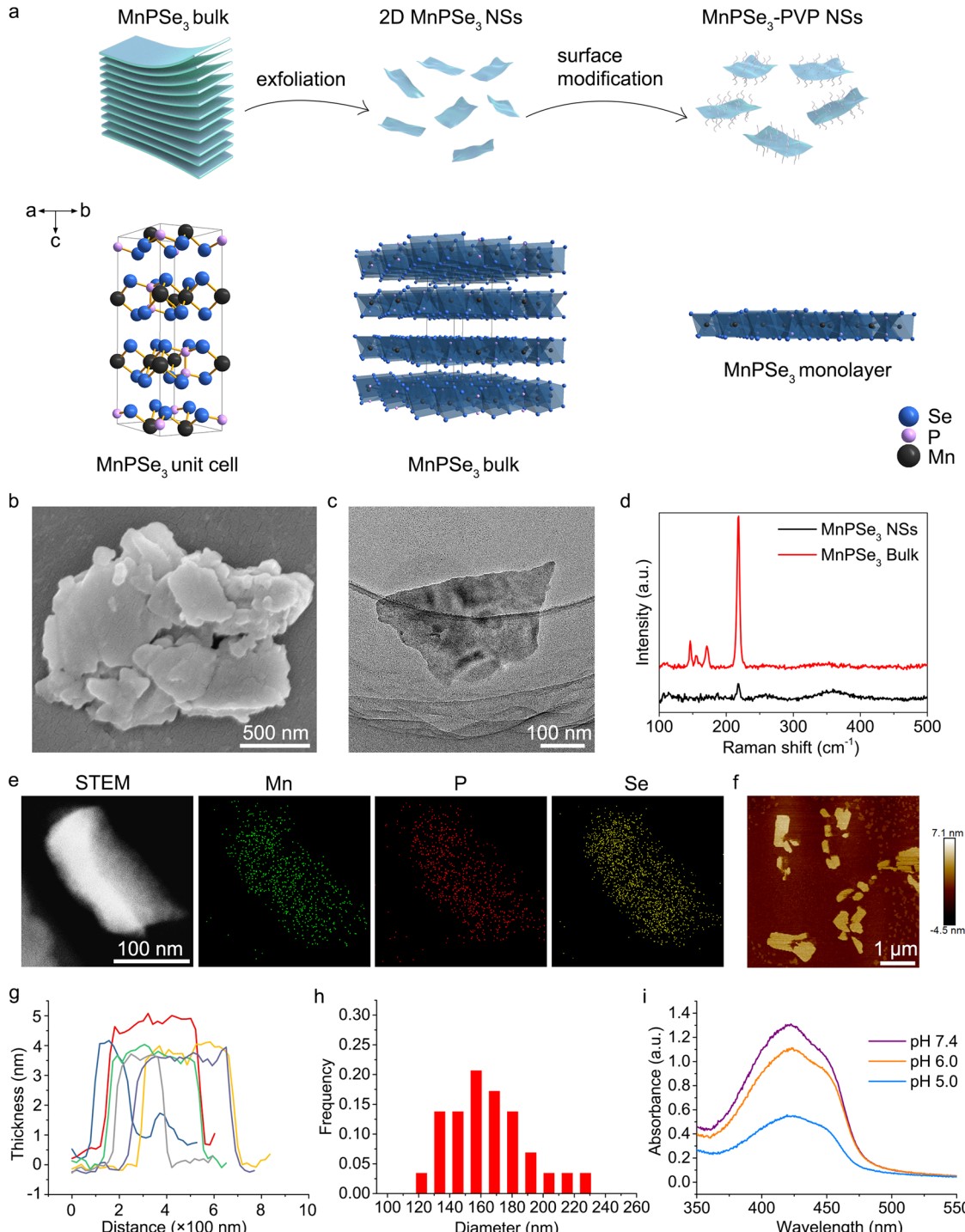

**Fig. 2 | MPS preparation and characterization. a** Schematic illustrations of the synthesis of MPS and its molecular structure. **b** SEM image of bulk MPS. Scale bar, 500 nm. **c** TEM image of MPS NSs. Scale bar, 100 nm. **d** Raman spectra of bulk MPS and MPS NSs. **e** Elemental mapping of MPS NSs. Scale bar, 100 nm. **f** Typical AFM images of MPS NSs. The experiments in **b**, **c**, **e**, **f** were repeated independently three times with similar results. **g**, **h** Thickness and lateral size distribution analysis of MPS NSs. **i** Catalytic oxidation of DPBF by MPS NSs in the conditions of different pHs. Source data are provided as a Source Data file.

## Immune-stimulatory MPS-PVP boosts the immune responses of RAW264.7 macrophages

Macrophages are indispensable for effective innate and adaptive immune responses against biofilm[10]. To test whether the changes of immune-inhibitory and co-activating molecules induced by MPS stimulates subsequent immune responses, the polarization status and phagocytosis capability of RAW264.7 macrophages were profiled. Biofilm converts macrophage polarization from the pro-inflammatory phenotype (M1) to the anti-inflammatory phenotype (M2), while the latter features the overly production of anti-inflammatory cytokines. Immunofluorescent staining and flow cytometric analysis demonstrate a significantly larger number of M1 phenotype macrophages in MPS-PVP groups, indicating a less exhausted state of macrophages (Fig. 5a–c and Supplementary Fig. S5). Besides, the MPS-PVP groups also demonstrate the elevated releases of pro-inflammatory cytokines (Fig. 5d). The release of pro-inflammatory cytokines is associated with enhanced antigen priming and subsequent presentation by APCs, which would further amplify adaptive anti-biofilm immune responses.

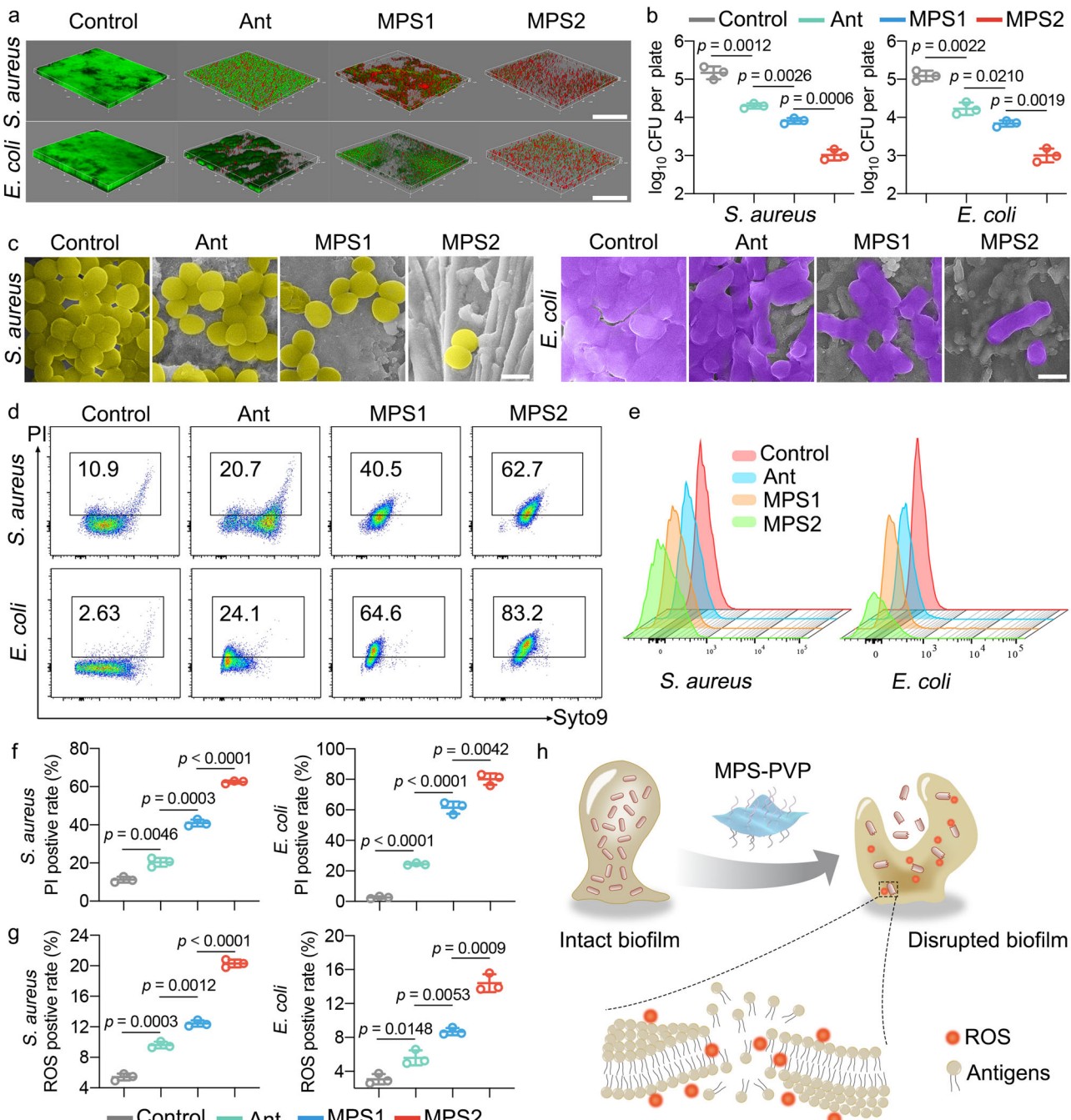

**Fig. 3 | MPS-PVP leverages biofilm as an endogenous antigen deposit. a** 3D confocal micrographs of biofilm (live bacteria, green; dead bacteria, red). Scale bar, 100 μm. **b** quantitative analysis of biofilm biomass ($\log_{10}$ CFU per titanium plate). **c** SEM images of biofilm. To better visualize the biofilm, *S. aureus* were pseudo-colored yellow, while *E. coli* were pseudo-colored purple. Scale bar, 1 μm. **d**, **f** Representative flow cytometry plots and quantitative analysis of Syto9/PI staining on extracted biofilm. **e**, **g** Flow cytometric analysis of DCFH-DA stained biofilm for the detection of ROS. **h** Schematics of MPS-PVP for enhancing biofilm antigen expression. The experiments in **a** and **c** were repeated independently three times with similar results. **b**, **f**, **g** Data are mean ± s.d. ($n = 3$ per group) and $n$ represents biologically independent experiments. Two-tailed, unpaired *t*-test, exact *p*-values. Ant, antibiotics. Source data are provided as a Source Data file.

Further investigations evidence that MPS-PVP-treated RAW264.7 macrophages give a higher bacterial phagocytosis rate than other groups (Fig. 5e–g). Considering that vancomycin as the classical antibiotic used for biofilm infection featured limited immune-stimulatory impacts, such state-of-art inorganic immune-stimulatory strategy provides an effective tool for reversing the immunosuppressive biofilm microenvironment in alternative to the conventional antibiotic therapy (Fig. 5h).

To test whether PVP in the MPS-PVP influences the immune response of macrophages, flow cytometry analysis was performed on RAW264.7 macrophages incubated with different concentrations of PVPs. As shown in Supplementary Fig. S6, addition of PVPs did not change the phenotype of macrophages. This result indicates that MPS, rather than PVP, is critical to provoke the macrophage polarization.

## Immune-stimulatory MPS-PVP inflames anti-biofilm humoral immunity in vivo

Given the immune-stimulatory capacity of MPS-PVP in vitro, we generated implant infection mouse model to assess the in vivo

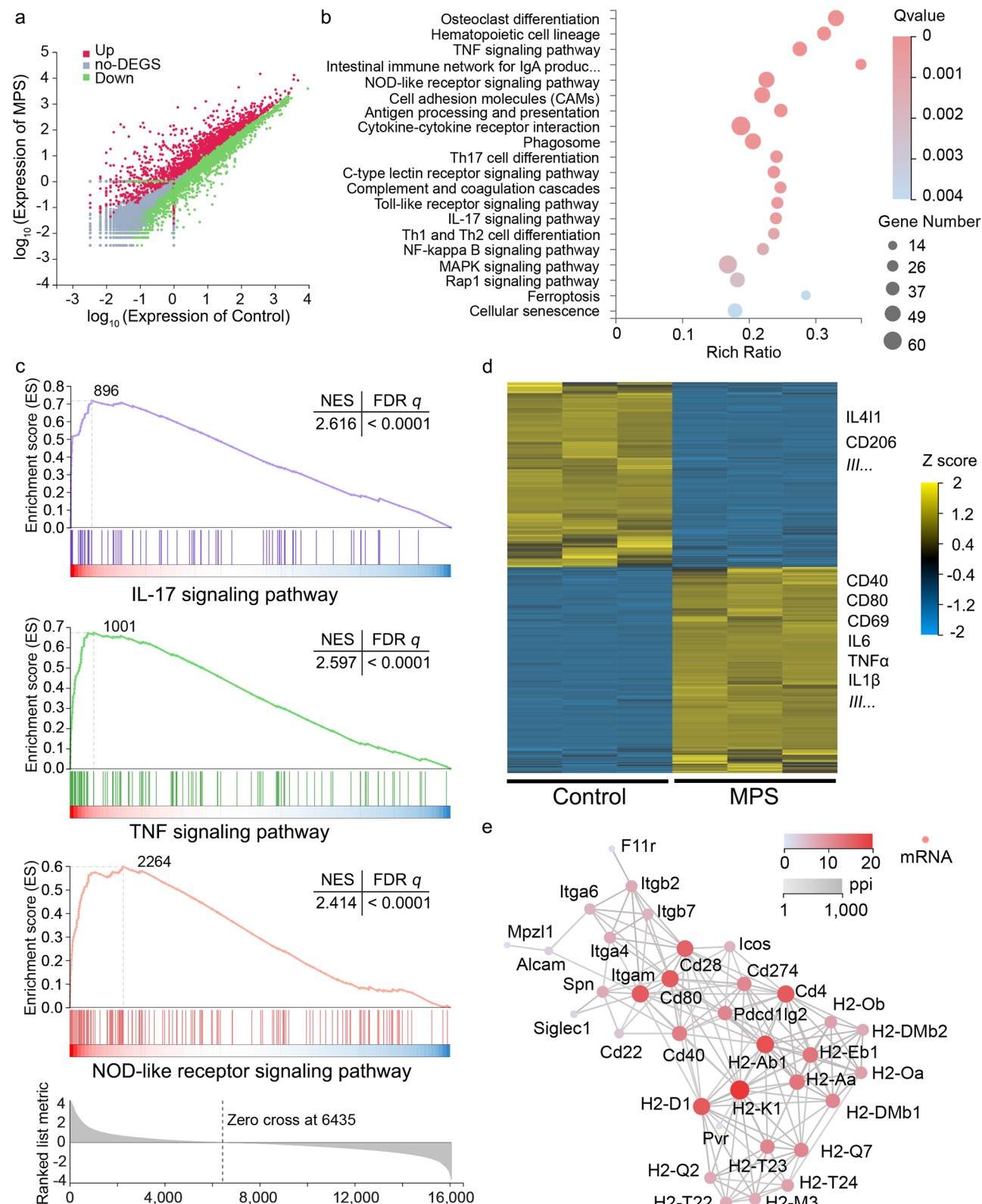

**Fig. 4 | MPS-PVP activates co-stimulatory pathways in RAW264.7 macrophages.** **a** Volcano plots showing the expression levels of genes in different groups. **b** KEGG enrichment analysis reveals top 20 statistically significant (Q values <0.05) biological processes regulated by MPS-PVP. **c** Gene set enrichment analysis (GSEA) of representative immune signaling pathways in macrophages after MPS-PVP treatment. The ticks below the line represent the rank of each gene. NES, normalized enrichment score. **d** Heatmap showing DEGs after MPS-PVP treatment; yellow, upregulation; blue, downregulation; |log₂ fold change| ≥ 1, *Q* values < 0.05. Key DEGs are marked on the right. **e** Gene–gene interaction network for genes associated with cell adhesion molecules in macrophages. The network nodes and the lines correspond to each gene and gene-gene interactions, respectively.

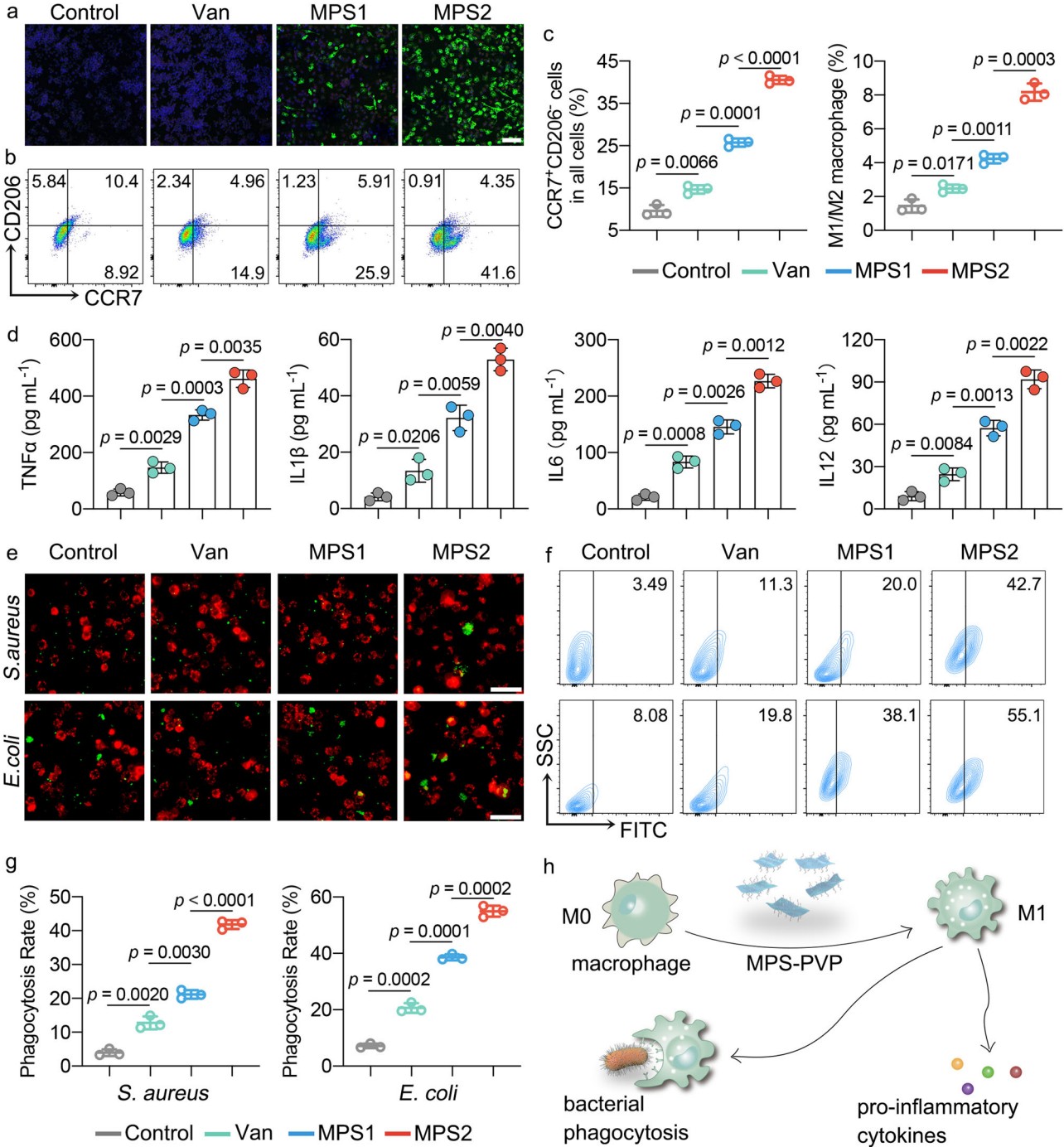

**Fig. 5 | Immune-stimulatory MPS-PVP facilitates antigen processing and presentation of RAW264.7 macrophages. a** Immunofluorescence staining of RAW264.7 macrophages for iNOS (green), CD206 (red), and DAPI (blue). Scale bar, 100 μm. **b, c** Flow cytometry analysis displaying changes in CCR7 (M1 marker) and CD206 (M2 marker) expression in RAW264.7 macrophages after different treatments. M1/M2 macrophage equals to CCR7 positive/CD206 positive cells. **d** Concentrations of TNFα, IL1β, IL6, and IL12 in the supernatants of RAW264.7 macrophages after indicated treatments. **e** Fluorescence staining showing the bacterial phagocytosis 60 min after RAW264.7 macrophages (red) and bacteria (green) inoculation. Scale bar, 50 μm. **f, g** Representative flow cytometry plots (**f**) and quantification (**g**) of phagocytic clearance of bacteria by RAW264.7 macrophages with indicated treatments. **h** Schematics of MPS-PVP maintaining proinflammatory phenotypes of macrophages. The experiments in **a** and **e** were repeated independently three times with similar results. **c, d, g** Data are mean ± s.d. ($n = 3$ per group) and $n$ represents biologically independent experiments. Two-tailed, unpaired $t$-test, exact $p$-values. Van, vancomycin. Source data are provided as a Source Data file.

impact of MPS-PVP on anti-bacterial biofilm humoral immunity (Fig. 6a). *Staphylococcus aureus* is the main causative bacteria in implant infections, and nearly half of *Staphylococcus aureus* clinical isolates are methicillin resistant (methicillin-resistant *Staphylococcus aureus*, MRSA) in the United States[37–39]. Therefore, we chose a typical MRSA isolate (ATCC 43300) for the construction of implant infection

model in vivo. Vancomycin remains the standard of care for MRSA implant infections in clinical practice currently[40], and vancomycin is commonly used as positive control for anti-MRSA nanomaterial studies[41,42]. On this ground, we chose vancomycin as clinically relevant control group to evaluate the antibacterial and anti-biofilm effect of MPS-PVP in vivo. The administration of MPS-PVP compared

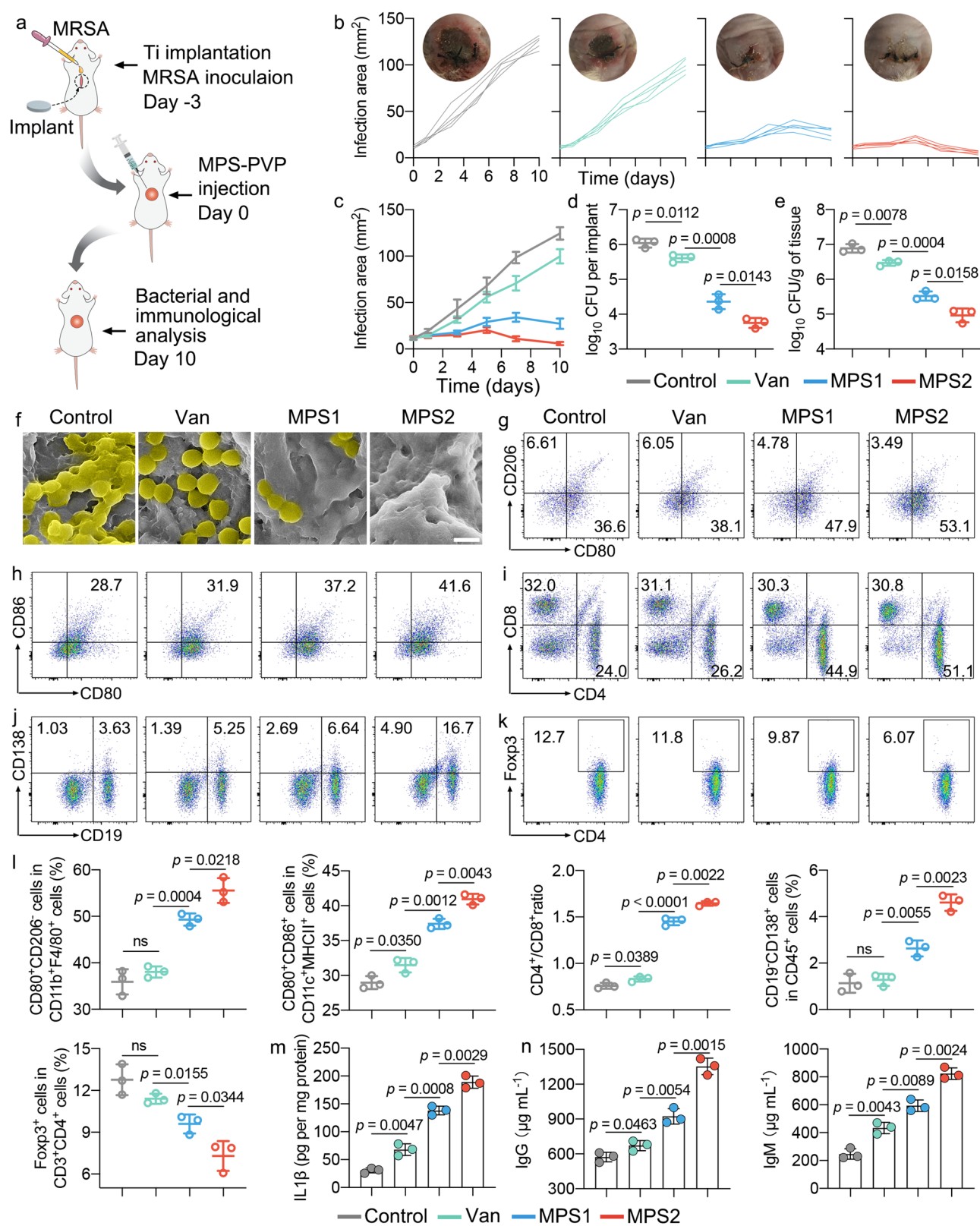

to the use of vancomycin resulted in significantly slower infection progression in all time intervals at the infection area assessments (Fig. 6b, c), which is consistent with bacterial count analysis (Fig. 6d, e). More importantly, the injected MPS-PVP demonstrates an attractive biofilm eradication benefit in comparison to using vancomycin, thus overcoming the drug-tolerance of implant-related biofilm in vivo[43] (Fig. 6f). Immune profiling of infection draining lymph nodes

(IDLN) demonstrates the MPS-PVP activated multiple immune cells in the pathway of humoral immunity (Fig. 6g–l and Supplementary Figs. S7–12). MPS-PVP treatment significantly elevated the fraction of pro-inflammatory M1-like macrophages (CD11b[+]F4/80[+]CD80[+]CD 206[-]), which overexpressed CD80. Similarly, MPS-PVP generated a pronounced increase of mature DCs (CD11c[+]MHC-II[+]CD80[+]CD86[+]) in the IME. Due to the formation of biofilm, APCs are in frustrated and

**Fig. 6 | Immune-stimulatory MPS-PVP permits biofilm control dependent on humoral immunity. a** Scheme showing the preparation of "in situ" implant MRSA infection model and therapeutic treatment of MPS-PVP. **b, c** Individual infection area growth curves and representative photos (**b**) and average infection area growth curves (**c**) of infected mice after indicated treatments. (*n* = 5 biologically independent mice. Data are mean ± s.d.). **d, e** Bacterial CFU counts of extracted implants (**d**) and peri-implant tissues (**e**) of infected mice. **f** Representative SEM images of biofilm formation in vivo. To better visualize the biofilm, *S. aureus* were pseudo-colored yellow. Scale bar, 1 μm. The experiments in **f** were repeated independently three times with similar results. **g–k** Representative flow cytometry plots of M1-phenotype macrophages (CD80$^+$CD206$^-$) by gating on CD11b$^+$F4/80$^+$ cells

(**g**), mature DCs (CD80$^+$CD86$^+$) by gating on CD11c$^+$ MHC-II$^+$ cells (**h**), CD4$^+$ T cells (CD4$^+$) by gating on CD3$^+$ T cells (**i**), plasmablasts (CD138$^+$CD19$^+$) and plasma cells (CD138$^+$CD19$^-$) after gating on CD45$^+$ cells (**j**) and T$_{reg}$s (Foxp3$^+$) by gating on CD3$^+$CD4$^+$ T cells (**k**) in dissected IDLNs. **l** Quantification analysis of corresponding immune cell populations in **g–k**. **m** Cytokine expression profiles in the infected tissues. **n** Serum antibody levels in mice with different treatments. **d, e, l, m, n** Data are mean ± s.d. (*n* = 3 per group) and *n* represents biologically independent experiments. Two-tailed, unpaired *t*-test, exact *p*-values. ns, not significant. Van, vancomycin. Note, vancomycin and MPS-PVP were administrated through local injection in the infected area. Source data are provided as a Source Data file.

exhausted state featured by anti-inflammatory polarization of macrophages and immaturation of DCs. The status transitions of APCs suggested MPS-PVP facilitated bacterial antigen processing and presentation. With higher infiltration of CD4$^+$ T cells, comes stronger humoral immunity, and the inhibition in CD4$^+$ T cell priming limits biofilm eradication[44]. As expected, a significant higher levels of CD4$^+$ T cell (CD3$^+$CD4$^+$) infiltration was observed in MPS-PVP treated mice. When infection occurs, antigen-primed B lymphocytes clonally expand and carry secondary diversification of the immunoglobulin genes. The rare winner cells are then selected, naming plasma cells and memory B cells which are the effector cells to produce bacteria-specific neutralizing antibodies in humoral immunity[45]. MPS treatment resulted in significant enrichment of plasma cells (CD138$^+$CD19$^-$) and plasmablasts (CD138$^+$CD19$^+$) in the IME, indicating the antigen-specific humoral immune responses induced by MPS-PVP. Besides, in groups treated with MPS-PVP, we observed a sharply reduction in T$_{reg}$s (CD4$^+$Foxp3$^+$) infiltrates, which suggests that MPS-PVP was restricting the migration of T$_{reg}$s into the IME. Immunologically cold biofilms are characterized by a short of plasma B cells, instead being populated by suppressor cells such as T$_{reg}$s[13,44]. These results highlight the importance of MPS-PVP in changing the balance of inhibitory and activating immune cells to enhance the efficacy of humoral immunity. Notably, the mice treated with MPS-PVP exhibited substantial increase of pro-inflammatory cytokines in the IME such as TNFα, IL1β, IL6, and IL12 (Fig. 6m and Supplementary Fig. S13). These results further proved that MPS-PVP can elicit strong anti-biofilm immune responses in poorly immunogenic biofilm. In the resistant bacterial infections, the antibody response is deemed as the key correlate of protection and it's the basis for the neutralization of bacteria. However, the development of antibody production is greatly impaired in the presence of biofilm[4,13]. We observed a significant higher levels of serum IgG and IgM in the mice treated with MPS-PVP (Fig. 6n). Taken together, these analyses suggest that MPS-PVP inflames strong anti-biofilm antibody responses. In addition, we analyzed the maturation of DCs in the PVPs-treated mice which reflects the influences of PVPs on immune responses in vivo. As shown in Supplementary Fig. S14, administration of PVPs on the infected mice model does not induce the percentage fluctuations of mature DCs. This data evidence the pivotal role of MPS, rather than PVP, in eliciting antibacterial immune responses.

Besides, to evaluate whether the inflamed immune responses would lead to systematic side effects, pathological analysis of major organs was performed after MPS-PVP treatments. The results indicated that MPS-PVP did no harm to major organs in vivo (Supplementary Fig. S15). To understand whether bacterial antigens triggered by MPS-PVP treatment would influence normal microbial homeostasis or not, we investigated the composition of the gut microbiota in mice using 16S rRNA-targeted sequencing analysis[46,47]. As shown in Supplementary Fig. S16, MPS-PVP treatment exhibits a negligible effect on gut microbiome as inferred from the highly similar microbial compositions between MPS-PVP and PBS treatment groups. These results further prove the biosafety of MPS-PVP therapy.

### MPS-PVP boosts long-term humoral immunity for enhanced presurgical neoadjuvant immunotherapy

Surgical removal of infected implant is a stand-of-care procedure in clinic for chronic implant infections, followed by long-term antibiotic suppression therapy and subsequent new implant replacement[48]. However, infection recurred in a majority of patients after surgery due to the incomplete resection of infected targets and the presence of micrometastatic bacterial deposits, which is urgent to be tackled with[49]. Neoadjuvant immunotherapy is a concept used in cancer treatment, which aims to promote systemic immune responses and eradicate micrometastatic tumor deposits that otherwise would cause postsurgical relapse[50]. Here, we hypothesized that MPS-PVP neoadjuvant immunotherapy before surgical removal of infected implant can enhance systemic immune responses to eliminate micrometastatic bacterial deposits and boost humoral immune memory effect for preventing infection relapse. The in vivo protocol is shown in Fig. 7a. In control and vancomycin groups, infection relapsed after new implant replacement, as evidenced by the infection area measurement and bacterial count quantitative analysis (Fig. 7b–f). On the contrary, a significant decrease in infection area and the bacterial count was visualized in MPS-PVP groups. Furthermore, as shown in Fig. 7g, h, MPS-PVP treatment shows the lowest infiltration of neutrophils and bacteria. These results indicated that pathologic complete response was elicited by MPS-PVP neoadjuvant immunotherapy[41]. We further evaluated the immune profiles in this model. The treatment with MPS-PVP induced the highest levels of memory B cells (B$_{mem}$, IgG$^+$IgD$^-$) infiltrates (Fig. 7i, j and Supplementary Fig. S17). Notably, a significantly higher percentage of CD80$^+$ B$_{mem}$, which is a type of memory B cells that can differentiate into antibody-forming cells[51], was observed in MPS-PVP group (Fig. 7k, l). In addition, animals treated with MPS-PVP had increased serum pro-inflammatory cytokines release, compared with control and vancomycin groups (Fig. 7m and Supplementary Fig. S18). Notably, a significant higher levels of IgG was detected in the mice treated with MPS-PVP (Fig. 7n). Taken together, these results suggest that MPS-PVP neoadjuvant immunotherapy effectively elicited systemic immune responses and memory humoral immunity for the elimination of micrometastatic bacterial foci and prevention of infection relapse.

### Discussion

Igniting the immunologically cold biofilm remains a major challenge for biofilm eradication. Activation of key immune cells that drive anti-biofilm humoral immune responses, including APCs, CD4$^+$ T cells and B cells, is an attractive strategy to achieve this goal. The immune system is spatiotemporally influenced by biofilm, so therapies that target the immune suppressive pathways should be controlled spatiotemporally as well, to maximize the therapeutic index. The discovery of the intrinsic immunomodulatory effects of nanomaterials permits us to shape the immune responses toward the desired directions. They can be highly multivalent for a complex engagement in the host immune system[16]. Herein, we synthesized a 2D inorganic MPS nanosheets and investigated its potential to stimulate anti-bacterial biofilm humoral immunity for the treatment of implant infections.

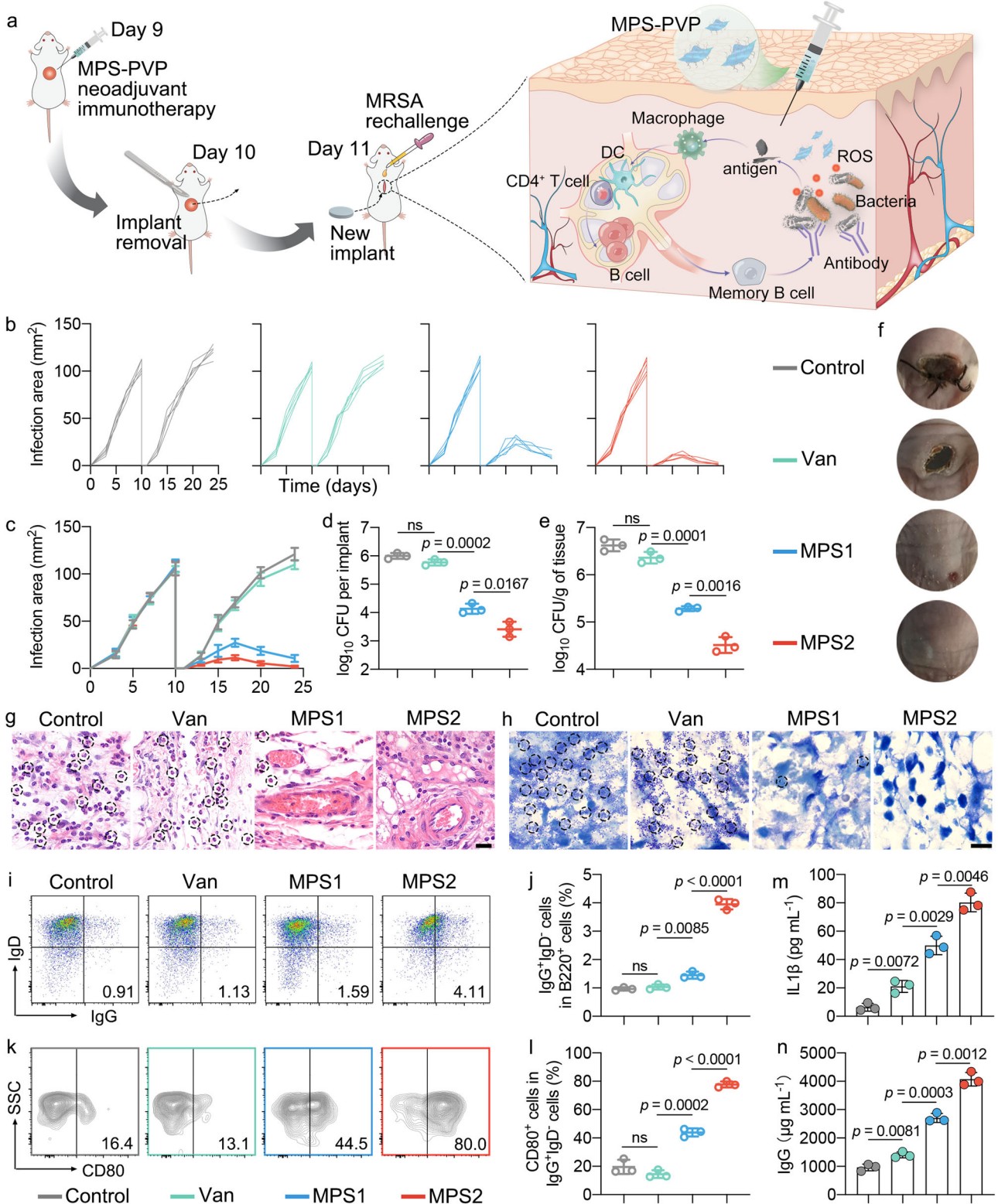

MPS nanosheets consists of Mn, Se and P elements which all play essential roles in immunological activities. In previous studies, it was shown that $Mn^{2+}$ could boost immune response as STING (stimulator of IFN genes) agonist to facilitate antigen uptake and presentation, germinal center formation, B cell responses and antibody production[52–54]. Colloidal Mn salt showed good adjuvant effects for T cell-dependent antigens, such as ovalbumin, and T cell independent antigens, such as bacterial capsular polysaccharides[54]. Researchers also make use of the immune-stimulatory effect of $Mn^{2+}$ to induce DCs

maturation and elicit pathogen-specific memory B cell response using manganese dioxide nanoparticles in *S. aureus* infection mice model[55]. In terms of the role of Se element, a previous study showed that selenium supplementation promoted glutathione peroxidase 4 (GPX4) expression in T cells, amplified the number of follicular helper T cells and boosted antibody responses in mice and young adults immunized with influenza vaccination[56]. In a *Vibrio cholerae* whole-cell vaccine mouse model, administration of selenium nanoparticle resulted in robust *Vibrio. cholerae*-specific IgG and IgA antibody responses in

**Fig. 7 | Pre-surgical neoadjuvant immunotherapy confers long-term humoral immune protection for preventing infection relapse. a** Schematics for the pre-surgical neoadjuvant immunotherapy that prevents postsurgical MRSA infection relapse. In 9 days after the establishment of implant infection model, mice were treated with MPS-PVP followed by the implant removal surgery on day 10. New implants were then inserted with fresh MRSA inoculation on day 11. **b, c, f** Individual infection area growth curves (**b**), average infection area growth curves (**c**) and representative photos (**f**) of mice after indicated treatments. (n = 5 biologically independent mice. Data are mean ± s.d.). **d, e** Bacterial CFU counts of extracted implants (**d**) and peri-implant tissues (**e**) in the pre-surgical neoadjuvant immunotherapy model. **g** Representative images of HE staining of tissues in the infected area. The circles indicate neutrophils which are associated with the infection severity. Scale bar, 20 μm. **h** Representative images of tissue sections analyzed for bacterial biomass using Giemsa staining. *S. aureus* are indicated by circles. Scale bar, 10 μm. **i, j** Representative flow dot plots (**i**) and statistical analysis (**j**) of memory B cells ($B_{mem}$, IgG$^+$IgD$^-$) by gating on B220$^+$ cells. **k, l** Representative flow dot plots (**k**) and statistical analysis (**l**) of CD80$^+$ $B_{mem}$ which can differentiate rapidly into antibody-forming cells by gating on IgG$^+$IgD$^-$ cells. **m** Levels of IL1β cytokine in the serum. **n** Serum IgG antibody levels in different groups. **d, e, j, l, m, n** Data are mean ± s.d. (n = 3 per group) and n represents biologically independent experiments. Two-tailed, unpaired t-test, exact p-values. ns, not significant. Van, vancomycin. Note, vancomycin and MPS-PVP were administrated through local injection in the infected area. Source data are provided as a Source Data file.

serum and saliva[57]. As for P element, in a mouse melanoma model, it was demonstrated that black phosphorous nanosheet could act as immune-potentiating nanoadjuvant which elicited intense antitumor immune responses and markedly amplified checkpoint blockade effects[58].

To evaluate the antibacterial and immunomodulatory effect of MPS-PVP in vivo, we constructed mouse model of MRSA implant infection. Vancomycin is widely used in clinical practice for the management of MRSA implant infection[40], and vancomycin treatment is often used as control group to evaluate the anti-MRSA effect of nanomaterials[41,59]. Therefore, vancomycin treatment is selected as the control group in our in vivo studies. The tissue penetration efficacy and theraputic effect of intravenous administrated vancomycin is limited in implant infections, while direct application of vancomycin to the infected area can yield better infection control[39,60]. Thus, local injection of vancomycin and MPS-PVP was applied for our in vivo experiments.

APCs, which mainly consist of macrophages and DCs, play a crucial role in the processing of endogenous antigens and the initiation of anti-bacterial adaptive immunity[61,62]. In normal host defense, M1 phenotype macrophages are the dominant cell population which induce pro-inflammatory immune responses, phagocytose bacteria, release antimicrobial agents, and present bacterial antigens[63]. Mature DCs also play an essential role in processing and presenting bacterial antigens to T cell receptors (TCR) for T cell activation[62,64], and the enhancement in DCs maturation with biomaterials could facilitate the activation of anti-bacterial adaptive immune responses[65]. However, in implant infection, biofilm polarizes the immune cells toward an anti-inflammatory phenotype featured by macrophage M2 phenotype transition with impaired antigen processing and presentation[2]. In addition, infections increase the expression of immune-inhibitory molecules while decrease the expression of positive costimulatory molecules in APCs, which would suppress T cell function by binding with the receptor expressed by T cells[14].

In current study, MPS-PVP triggered more efficient biofilm antigen release in a concentration-dependent manner than conventional antibiotics. The resulting antigens can serve as endogenous antigen deposits for subsequent APC priming and B cell response. Essentially, the higher biofilm bacterial antigen load present in the context of immunotherapy will hypothetically lead to the priming of more bacteria-specific antibodies circulating systemically. Through RNA sequencing of RAW264.7 macrophages, we found that MPS-PVP strongly activated anti-bacteria pathways such as NOD-like receptor, TLR, and IL17 signaling pathways. Besides, MPS-PVP downregulated the expression of immune-inhibitory molecules, while upregulated the expression of co-activating molecules in macrophages, which is vital for bacterial antigen presentations and subsequent T cell priming[66,67]. In vitro data showed that MPS-PVP enabled RAW264.7 macrophages with pro-inflammatory phenotype transition and enhanced bacterial phagocytosis. In vivo flow cytometry analysis revealed that MPS-PVP promoted both macrophages M1 phenotype polarization and DCs maturation, which is essential for the presentation of bacterial antigens to T cells and B cells, thereby initiating adaptive immune responses[59].

Protective immunity against bacteria is driven mainly by antibodies produced by B cells, which is called humoral immunity. The priming and activation of B cells require the antigens processed by APCs and co-stimulatory signals from CD4$^+$ T cells[68]. B cells recognize bacterial antigens by B-cell receptor (BCR), the activated B cells then enter germinal centers to undergo somatic hyper-mutation and affinity maturation of their BCR. Those cells further differentiate into either long-lived plasma cells that constantly release antigen-specific antibodies or memory B cells that take part in a recall response upon repeated exposure to the antigen[17]. Although humoral immunity plays an indispensable role in biofilm elimination[5], there are rare reports on humoral immunity modulation using nanomaterials for biofilm management. Our study utilizes endogenous biofilm antigens for the activation of humoral immunity, which outperforms conventional vaccines with limited antigen epitopes. In situ implant infection model reveals a robust humoral immunity activation and anti-biofilm efficacy upon injection of MPS-PVP. The phenotype transitions of APCs, T cells, and plasma B cells in the IDLN as well as the cytokines and antibodies in the blood evidence the notable immune-stimulatory effect of MPS in vivo.

Neoadjuvant immunotherapy as an adjuvant to the surgical removal of tumor has yielded encouraging response rates with improved long-term survival rates in clinical practice. The surgical treatment of implant infection resembles tumor resection with possible residual focus beyond the resected area, which are the source of postsurgical relapse[69]. Besides, the lack of immune memory responses will lead to less effective immune activation upon postsurgical bacterial rechallenges, which further increases the risk of infection relapse[70]. Enhanced postsurgical infection control by MPS treatment is associated with the generation of more bacteria-specific memory B cells in IDLN as well as promoted bacteria-specific IgG and cytokine production, indicating the functioning of systemic anti-biofilm memory response.

Essentially, the high biofilm antigen load and co-activator signals stimulated by MPS in the context of neoadjuvant therapy will result in the priming of enhanced amount of bacteria-specific plasma B cells and memory B cells circulating systemically, which facilitates the detection and elimination of residual bacterial deposits, as well as the clearance of rechallenged bacteria. Overall, this inorganic immune-stimulatory platform provides a more comprehensive humoral immunity activation and stronger biofilm eradication than conventional antibiotic therapy. This study would inspire continued innovation in designing nanomaterials featuring immune-stimulatory effects prior to the implementation of such a novel immunotherapeutic approach.

## Methods

### Mice

Six-to-eight-week-old BALB/c mice were purchased from Laboratory Animal Management Department, Shanghai Family Planning Research

Institute. All animal procedures complied with the institutional policies and federal guidelines by the Institutional Animal Care and Use Committee of the Shanghai Sixth People's Hospital. Experimental procedures were specifically approved by the ethics committee from Shanghai Sixth People's Hospital, officially registered as DWLL2021-0755. Mice were housed in temperature-controlled (22 ± 2 °C) facilities with 12 h dark-light cycles and an average humidity rate between 40% and 70%.

## Cells and bacteria

MC3T3-E1 osteoblast precursor cells and RAW264.7 macrophages were purchased from Shanghai Institute of Cell Biology, Chinese Academy of Science. *S. aureus* (ATCC43300) and *E. coli* (ATCC35218) were obtained from the American Type Culture Collection. The cells were cultured in Dulbecco's modified Eagle's medium (Gibco) containing 10% fetal bovine serum (Gibco). Bacteria were grown in trypticase soy broth (TSB, Solarbio, China). The cells and bacteria were passaged every 1 to 2 days.

## Synthesis of MPS NSs

Bulk MnPSe$_3$ crystals were purchased from Mknano (Nanjing, China). Bulk MnPSe$_3$ (2 mg mL$^{-1}$) in the solution of N-methyl-2-pyrrolidone (Aladdin, >99.0%) was treated with probe sonication in an ice bath for 24 h with a power of 600 W. The probe sonicated for 5 s with an interval of 3 s. The obtained solution was centrifuged (cence, H2500R-2, China) for 10 min at 4000 rpm to remove large bulks. The resulting supernatant solution was collected and further centrifuged for 30 min at 14,000 rpm. Then, MPS NSs were collected after centrifugation and washed with ethanol and water. For biomedical applications, the MPS NSs were further modified with polyvinylpyrrolidone (PVP). Briefly, 25 mg of MPS NSs and 50 mg of PVP (Mw = 8000, Aladdin) were dispersed in 50 mL of ethanol and refluxed at 50 °C for 24 h. Excess PVP was removed by centrifugation and washing with ethanol and water. MPS-PVP was then obtained for subsequent uses.

## Characterization

SEM images were obtained on a field-emission Hitachi SU-8000 microscope. TEM images, elemental mappings and linear scanning were conducted on a JEM-2100F transmission electron microscope. Raman spectra were acquired on a high-resolution Raman spectrometer (HORIBA LabRAM HR800). Atomic force microscopy (AFM) images were obtained with Bruker Dimension Icon. The thickness of MPS NSs was acquired through analyzing AFM data with the software of Nano Scope Analysis (Bruker). The lateral sizes of MPS NSs were analyzed with the software of Nano Measure using TEM images.

## Detection of ROS in vitro

Diphenylisobenzofuran (DPBF, Sigma, 200 μM) and H$_2$O$_2$ (Sigma, 100 μM) were added into the solution of MPS-PVP NSs (10 μg mL$^{-1}$) with various pH values (7.4, 6.0 and 5.0) for 12 h. The absorbance of DPBF at 420 nm was then evaluated with a UV–vis–NIR spectrometer (UV-3600, Shimadzu).

## CCK-8 tests

Biocompatibility of MPS-PVP was assessed with CCK-8 test using MC3T3-E1 cells. In brief, cells (1 × 10$^4$ per well) were seeded in 96-well plates and incubated overnight. Afterwards, the cells were treated with MPS-PVP NSs at various concentrations for 48 h. The culture medium was then replaced with CCK-8 solution, and incubated for 90 min. Absorbance measurements were conducted at the wavelength of 450 nm using a microplate reader (Bio-TekELx800, USA). Cell viability was normalized to the control group.

## Biofilm antigen release tests

Before tests, the frozen cultures of *S. aureus* and *E. coli* were incubated in TSB overnight. After two consecutive transfers in TSB, the concentration of activated bacteria was adjusted to the 0.5 McFarland standard, which corresponds to 10$^8$ Colony Forming Units (CFU) mL$^{-1}$. The bacterial solution was further diluted to 10$^6$ mL$^{-1}$, then added into a 24-well plate containing sterile titanium plates (diameter, 1 cm). After incubation for 24 h at 37 °C, the plates were taken out and washed with PBS thrice, in order to remove planktonic bacteria. The titanium plates with biofilm were then transferred into a new 24-well plate. 1 mL of TSB, 1 mL of antibiotic solution (vancomycin for *S. aureus*, ofloxacin for *E. coli*, 20 μg mL$^{-1}$), 1 mL of MPS-PVP solution (10 μg mL$^{-1}$) and 1 mL of MPS-PVP solution (20 μg mL$^{-1}$) were added into the plates (denoted as Control, Ant, MPS1 and MPS2, respectively.) After incubation for another 24 h at 37 °C, the titanium plates were retrieved, and washed with PBS to remove floating bacteria. To visualize the biofilm on the titanium plates, the biofilm was stained with Syto9/PI staining kits (Invitrogen) for 15 min in the dark. Then, the plates were washed and photographed with a TCS SP8 laser scanning confocal microscope (Leica, Germany) using a HC PL APO CS 40×/0.85 dry objective. For Syto9 fluorescence, a 488 nm laser line was used, and a PMT 2 detector was used at a range of 493-546 nm. For PI fluorescence, a 552 nm laser line was used, and a HyD 3 detector was used at a range of 557–627 nm. z-stacks were generated with ~1 μm intervals; Obtained z-stacks were rendered into three-dimensional (3D) projections with LAS X software (Leica). For SEM observation, the biofilm was fixed with glutaraldehyde at 4 °C overnight, then dehydrated with a series of ethanol at the concentrations of 50%, 70%, 80%, 90%, and 100%. The biofilm was then treated with vacuum frozen and spray-gold. Finally, the biofilm was observed with scanning electron microscope (Hitachi SU-8000) using 10 kV accelerating voltage at 5 k and 20 k magnification (for *S. aureus*, working distance = 8.5 mm; for *E. coli*, working distance = 8.4 mm). To quantify the biofilm on the titanium plates, the biofilm was detached from the plates by ultrasonic concussion for 10 min and vortexing for 1 min. Next, 10-fold serial dilutions of the resulting suspension were made, and 100 μL of each dilution was spread onto the sheep blood agar plate, then incubated for 24 h at 37 °C and enumerated. To evaluate the degree of bacterial apoptosis and membrane damage, the detached bacterial solution (1 mL) was stained with 1 μL of Syto9 and 1 μL of PI staining for 15 min in the dark at room temperature. After centrifugation (Centrifuge 5418R, Eppendorf) at 6000 rpm for 3 min and washing with PBS, the bacterial solution was then collected for flow cytometry analysis. To measure the intracellular ROS level, the detached bacterial solution (1 mL) was incubated with 10 μM of DCFH-DA at 37 °C for 20 min. The stained bacteria were washed with PBS thrice, then analyzed with flow cytometer.

   To explore the actual cause of bacterial death, we conducted DNA degradation and lipid peroxidation assay. For DNA degradation test, we extracted the genomic DNA of *S. aureus* and *E. coli* with TaKaRa MiniBEST Bacteria Genomic DNA Extraction Kit Ver.3.0 (TaKaRa, Japan). Then, the genomic DNA was exposed to PBS, antibiotics (vancomycin for *S. aureus*, Ofloxacin for *E. coli*, 20 μg mL$^{-1}$), MPS1 (MPS-PVP, 10 μg mL$^{-1}$) and MPS2 (MPS-PVP, 20 μg mL$^{-1}$) in aqueous solution (pH 5.0, H$_2$O$_2$ 100 μM) at 37 °C for 12 h. The supernatant containing genomic DNA was then harvested by centrifugation at 12,000 rpm for 5 min. After 1% TAE agarose gel electrophoresis and GelRed staining, DNA cleavage products were imaged with Bio-Rad FX imaging system (Bio-Rad, USA). For lipid peroxidation assay, the detached bacteria from the biofilm on the titanium plates after various treatments was lysed with an ultrasonic homogenizer (SCIENTZ-IID, SCIENTZ, China) at 4 °C for 30 min with a power of 600 W. The probe sonicated for 3 s with an interval of 3 s. The lysate was centrifuged (GENESPED 1730R, Gene Company Limited) at 15,000 rpm for 15 min, the supernatant was

then collected for MDA measurement using Lipid Peroxidation MDA Assay Kit (Beyotime, China).

## RNA-seq sample preparation and data analysis

RAW264.7 macrophages were seeded in six-well plates at the density of $1 \times 10^5$ cells per well, and incubated at 37 °C at the atmosphere of 5% $CO_2$ for 24 h. Afterwards, the culture medium was replaced with fresh media or fresh media containing 20 µg mL$^{-1}$ of MPS-PVP. After incubation for another 24 h, the culture medium was discarded and washed with PBS thrice. 1 mL of Trizol (Invitrogen) was added to extract RNA, then frozen in liquid nitrogen immediately. The samples were then sequenced with DNBseq Platform in the company of BGI (Shenzhen, China). The collected data were analyzed with the software of Dr.Tom (BGI).

## Macrophage functional tests in vitro

RAW264.7 macrophages ($5 \times 10^4$ per well) were seeded in 24-well pates and incubated overnight. Then, the culture medium was discarded and replaced with either fresh medium, or fresh medium containing 20 µg mL$^{-1}$ of vancomycin, or fresh medium containing 10 µg mL$^{-1}$ of MPS-PVP, or fresh medium containing 20 µg mL$^{-1}$ of MPS-PVP (denoted as Control, Van, MPS1, and MPS2, respectively). After incubation overnight, the cells were washed with PBS and fixed in 4% paraformaldehyde. Next, the cells were sequentially permeabilized with 0.1% Triton-X for 30 min and blocked by 3% BSA for 1 h. Then, the cells were incubated with mouse monoclonal to iNOS (1:200, Abcam. Cat.# ab49999) and rabbit polyclonal to mannose receptor (1:200, Abcam. Cat.# ab64693) at 4 °C overnight. Donkey anti-mouse IgG H&L Alexa Fluor® 488 (1:200, Abcam. Cat.# ab150105) and donkey anti-rabbit IgG H&L Alexa Fluor® 594 (1:200, Abcam. Cat.# ab150076) were used to react with the primary antibodies in the dark. The cells were then stained with DAPI, and imaged with a fluorescence microscopy. For flow cytometry analysis, RAW264.7 macrophages with different treatments were harvested and resuspended in PBS. Single cells were firstly pretreated with anti-CD16/32 antibody (BioLegend. Cat.# 101320) for Fc receptors blockade. Afterwards, the cells were surface stained with APC-labeled anti-CCR7 (4B12, BioLegend. Cat.# 120108) for 30 min on the ice, then fixed and permeabilized with Cyto-Fast™ Fix/Perm Buffer Set (BioLegend). The cells were then intracellular stained with PE-labeled anti-CD206 (C068C2, BioLegend. Cat.# 141706). Besides, the supernatant was harvested to detect the cytokine levels of TNF-α (ANOGEN), IL-1β (ANOGEN), IL-6 (ANOGEN) and IL-12 (Dakewei Biotech) with enzyme-linked immunosorbent assay (ELISA) in accordance with the manufacturer's instructions.

To evaluate the phagocytic capability of macrophages, RAW264.7 cells with different treatments were first stained with Dil (Beyotime), then infected with CFDA-SE labeled *S. aureus or E. coli* at the multiplicity of infections (MOI) of 10:1 (bacteria/macrophages). 60 min after infection, the cells were washed with PBS to remove floating bacteria, then visualized with a fluorescence microscopy or analyzed with flow cytometer (LSR Fortessa, BD Biosciences).

To determine whether PVP effects the function of macrophages, RAW264.7 cells were cultured in either fresh medium, or fresh medium containing 10, 50 and 100 µg mL$^{-1}$ of PVP (denoted as Control, PVP-1, PVP-2, and PVP-3, respectively) for 24 h. The cells were then harvested for flow cytometry analysis as described above.

## "In situ" implant infection model

Male BALB/c mice (Six-to-eight-week-old, $n = 40$) undergone titanium plates implantation surgery as follows. Briefly, the mice were anaesthetized, shaved and disinfected. The back of the mice was dissected to expose subcutaneous layers in sterile environment. Subsequently, sterile titanium plates (diameter, 6 mm) were inserted and the wound was sutured. 100 µL of MRSA (ATCC43300, $10^6$ CFU mL$^{-1}$) was injected onto the implants subcutaneously. Three days later, the infected mice

were randomly allocated into four groups: Control, Van, MPS1, and MPS2. 100 µL of PBS (Control group), vancomycin (20 mg kg$^{-1}$, Van group), MPS-PVP (10 mg kg$^{-1}$, MPS1 group) or MPS-PVP (20 mg kg$^{-1}$, MPS2 group) were injected into the infection area of each mouse. The infection area of each mouse was monitored post-surgery. Ten days after different treatments, mice were euthanized using pentobarbital for bacterial analysis and flow cytometric analysis of infiltrated immune populations. The implants and surrounding tissues were dissected, weighed, and preserved on ice. Tissue samples were firstly homogenized with a tissue homogenizer. Single-cell bacterial suspensions were obtained through a combination of sonication and vortexing (10-min sonication and 30-s vortex, three cycle). The bacterial suspensions were serially diluted with PBS, plated on sheep blood agar plate, cultured at 37 °C overnight. The counting of CFU was performed manually. Bacterial counts were presented as CFU per implant or normalized to the weight of surrounding tissues and exhibited as CFU/g of tissue. In addition, the implants were prepared for SEM observation in a similar way to that described in the in vitro biofilm part. To assess the biosafety of MPS in vivo, major organs of mice were harvested as well, then sectioned and stained with hematoxylin-eosin (HE) staining.

To profile the immune cell composition in vivo, axillary lymph nodes (LN) were harvested immediately after the mice were euthanized. The LNs were mechanically dissected into small pieces, then digested with 1 mg mL$^{-1}$ collagenase IV and 100 µg mL$^{-1}$ deoxyribonuclease at 37 °C for 30 min. After filtered through a 70-µm filter, single cells were collected. Then, the single-cell suspension was treated with ACK lysis buffer (Gibco) for 5 min at room temperature to lyse RBCs. The single-cell suspension was incubated with anti-CD16/32 antibody for 10 min to block Fc receptors, and further incubated with fluorescent-labeled anti-mouse antibodies against CD45 (30-F11), CD11b (M1/70), F4/80 (BM8), CD80 (16–10A1), CD11c (N418), I-A/I-E (M5/114.15.2), CD86 (GL-1), CD3 (17A2), CD8a (53–6.7), CD4 (GK1.5), CD19 (6D5) and CD138 (281-2) for 30 min on the ice. For intracellular staining, the cells were fixed after surface staining and permeabilized with Cyto-Fast™ Fix/Perm Buffer Set or True-Nuclear™ Transcription Factor Buffer Set, then incubated with the antibody against CD206 (C068C2) or Foxp3 (MF-14), respectively. Multiparameter analyses were performed on an LSR Fortessa flow cytometer, and analyzed using FlowJo software (BD Bioscience). The detailed gating strategies were shown in the supplementary information. The information for staining antibodies and dilutions is provided in Supplementary Table 1.

To measure the cytokine secretion in vivo, the tissue homogenates were centrifuged for 5 min at 10,000 × $g$, supernatant was harvested and filtered with a 0.45-µm spin filter. Mouse cytokine kits of TNF-α (ANOGEN), IL-1β (ANOGEN), IL-6 (ANOGEN), and IL-12 (Dakewei Biotech) were used to determine the tissue concentrations of cytokines according to the manufacturer's protocols. Results were normalized to the total protein content of the sample which was measured using a bicinchoninic acid assay kit (Thermo). The plasma levels of IgG and IgM were also measured with ELISA per the manufacturer's instructions (Crystalchem).

To investigate the role of PVP in MPS-PVP on immune responses, "in situ" implant infection model was first established. Three days later, the mice were assigned into three groups randomly: Control, PVP-1, and PVP-2. 100 µL of PBS (Control group), PVP (10 mg kg$^{-1}$, PVP-1 group) or PVP (20 mg kg$^{-1}$, PVP-2 group) were injected into the infection area of each mouse. 10 days after different treatments, IDLN of mice were collected and flow cytometry analysis of immune cells were conducted as above mentioned.

## 16 S rRNA-targeted sequencing

Mice were treated with either PBS (Control group) or MPS-PVP (20 mg kg$^{-1}$, MPS group) after "in situ" implant infection model was established as above mentioned ($n = 12$ for each group). Ten days after

different treatments, the microbial community DNA was extracted from cecum content of mice using Magpure Stool DNA kit B (Magen, China) and amplified at the V3-V4 region of the 16S ribosomal RNA subunit gene. The validated libraries were sequenced on Illumina HiSeq platform (BGI, Shenzhen, China). Operational taxonomic unit (OTU) representative sequences were taxonomically classified by Ribosomal Database Project (RDP) Classifier v.2.2 with a minimum confidence threshold of 0.6, and trained on the Greengenes database v201305 using Quantitative Insights into Microbial Ecology (QIIME) v1.8.0[71]. Alpha and beta diversity were estimated using MOTHUR (v1.31.2)[72] and QIIME (v1.8.0) at the OTU level, respectively.

### Presurgical neoadjuvant immunotherapy model

Forty male BALB/c mice (Six-to-eight-week-old) were subjected to implantation and MRSA (ATCC 43300) inoculation to construct implant infection model on day 0. 9 days later, the mice were allocated into four groups randomly: Control, Van, MPS1, and MPS2. 100 μL of indicated formulation were injected into the infection area as previously described. On day 10, the mice were anesthetized and the back was surface-disinfected with iodophor. Under sterile circumstances, skin incision was made on the infected area, followed by implant retraction. The infected tissues were then debrided and removed thoroughly, and the wound was sutured. Mice were exposed to warming lamp and monitored until ambulatory. The next day, the mice were prepared for surgery as above. A new implant was inserted in the previously operating area, followed by fresh MRSA injection (100 μL of ATCC 43300 bacterial fluid at the concentration of $10^6$ CFU mL$^{-1}$) upon the new implant. Infection area growth was monitored daily by caliper measurements before the mice were euthanized using pentobarbital on day 24. Bacterial counts were performed in the same way as above described. The tissues surrounding the implant were dissected and fixed with paraformaldehyde, then submitted for paraffin embedding and sectioning. To visualize the infection, HE and Giemsa stainings were performed for the sectioned tissues. All stained slides were observed with light microscopy (Leica). IDLNs were processed for flow cytometry as above described. After Fc receptor block with anti-CD16/32 antibody, the cells were incubated with antibodies against B220 (RA3-6B2), IgD (11-26 c.2a), IgG (Poly4053) and CD80 (16–10A1). Samples were then acquired with the flow cytometer. The information for staining antibodies and dilutions is provided in Supplementary Table 2. Plasma cytokine and antibody levels were measured with ELISA per the manufacturer's protocols.

### Statistical analysis

All numeric data are presented as mean ± s.d. unless otherwise stated. Statistical analyses were conducted with two-tailed Student's $t$-test for comparison between two groups unless otherwise stated. One-way analysis of variance (ANOVA) with Tukey's post hoc test was used for multiple comparisons unless otherwise stated. All calculations and statistical analyses were conducted with Excel 2016 and GraphPad Prism 9 unless otherwise stated. A $p$-value of <0.05 was deemed statistically significant.

### Reporting summary

Further information on research design is available in the Nature Research Reporting Summary linked to this article.

## Data availability

The macrophage RNA-seq data have been deposited in NCBI Sequence Read Archive (SRA) database under the accession code PRJNA792491 The gut microbiota 16S rRNA-targeted sequencing data have been deposited in NCBI SRA database under the accession code PRJNA793390 The remaining data are available within the Article, Supplementary Information, or Source Data file. Source data are provided with this paper.

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

## Acknowledgements

We greatly acknowledge the support from the National Natural Science Foundation of China, grant no. 52002391 (to H.L.), grants no. 81772309 and 81974324 (to X.L.Z.), grant no. 81772364 (to H.S.), grant no. 21835007 (to J.L.S.), Key Research Program of Frontier Sciences, Chinese Academy of Sciences, grant no. ZDBS-LY-SLH029 (to J.L.S.), Shanghai Science and Technology Committee Rising-Star Program, grant no. 22QA1410200 (to H.L.), Shanghai Pilot Program for Basic Research-Chinese Academy of Science, Shanghai Branch, grant no. JCYJ-SHFY-2022-003 (to H.L.), Basic Research Program of Shanghai Municipal Government, grant no. 21JC1406000 (to J.L.S.), CAMS Innovation Fund for Medical Sciences, grant no. 2021-I2M-5-012 (to J.L.S.), Medical Guidance Scientific Research Support Project of Shanghai Science and Technology Commission, grant no. 19411962600 (to H.S.), and Experimental Animal Study Support Project of Shanghai Science and Technology Commission, grant no. 21140904800 (to H.S.).

## Author contributions

Conceptualization: C.Y., H.L., J.L.S., and X.L.Z. MPS preparation and characterization: H.L. and M.G. Bacterial experiments: C.Y., J.T., H.S., and Y.L. Cell and animal experiments: C.Y., H.L., and Y.L. Methodology: H.S. and Q.J.W. Writing–original draft: Y.C. and H.L. Writing–review, and editing: all authors. Supervision: J.L.S. and X.L.Z.

## Competing interests

The authors declare no competing interests.
