## [Peer Review File · Nature Communications]

Inorganic nanosheets facilitate humoral immunity against medical implant infections by modulating immune co-stimulatory pathwaysREVIEWER COMMENTS

Reviewer #1 (Remarks to the Author):

In this study, the authors used MPS to trigger ROS generation in biofilms and thus the release of bacterial antigens, which were then targeted by APCs to boost humoral immunity against biofilms. MPS was demonstrated as immune checkpoint blockade and helps reduce recurrence of infection after implant removal. Overall, this is a nice study with elegantly designed experiments and clear presentation. The findings advance the field with a new direction that is currently understudied. This reviewer only has a few minor comments.

- It was shown that MPS treatment generates ROS, but the actual cause of bacterial death was not demonstrated. More results/background will help understand the mechanism especially because the effects appear to be limited, i.e., 2 logs of killing at 200 ug/mL over 24 hours.
- The antigens released from bacteria are likely to be species non-specific. How would this affect the normal microbiome in the host?
- L44-46. This sentence is not clear and needs to be rephrased.
- L237. What is meant by "... implant biofilm invasive"?

Reviewer #2 (Remarks to the Author):

The manuscript by Yang and co-authors describes an inorganic nanosheet of MnPSe₃ as an a trigger of anti-biofilm humoral immunity.

Well written, up-to-date introduction, clear results description and high quality images
Few misspelling words: Giemsa (line 558 and 811); pH and PH (line 416); "the" instead of "he" (line 372)

Materials and methods section need some improvement:

1. please indicate were cells and bacteria cultures were described (line 393)
2. identify the type and model of centrifuge
3. for microscopy studies (confocal and SEM) describe the conditions used
4. line 536, please indicate the suppliers of the mouse cytokine kits

The inorganic nanosheets (MPS) were combined with a polymer (PVP) to ensure colloidal stability and biocompatibility. So, the cells and tissues are in contact with the MPS-PVP and this should be clear throughout the manuscript, starting with the title. Is there any information regarding the PVP important for the immune response?

RNA-seq reveals MPS-PVP as an essential tool for immune checkpoints in APCs, yet data was obtained only for macrophages. This should be stated in the text, and not only in the materials and methods section.

Response to reviewer I.

Comments from reviewer I:

In this study, the authors used MPS to trigger ROS generation in biofilms and thus the release of bacterial antigens, which were then targeted by APCs to boost humoral immunity against biofilms. MPS was demonstrated as immune checkpoint blockade and helps reduce recurrence of infection after implant removal. Overall, this is a nice study with elegantly designed experiments and clear presentation. The findings advance the field with a new direction that is currently understudied. This reviewer only has a few minor comments.

Response: Thank you very much for the constructive comments and kind recommendation. Please find the following detailed responses to your suggestions.

It was shown that MPS treatment generates ROS, but the actual cause of bacterial death was not demonstrated. More results/background will help understand the mechanism especially because the effects appear to be limited, i.e., 2 logs of killing at 200 ug/mL over 24 hours.

Response: We thank the reviewer for the constructive suggestions. We have performed additional background analysis which would help understand the bactericidal mechanisms and have provided related experimental results in our revised manuscript (**Figure S3-S4**):

According to previous studies, DNA degradation and lipid peroxidation are the immediate cause of bacterial death induced by ROS¹⁻³ (*Nat. Commun.* **12**, 745 (2021); *Nano Lett.* **19**, 7645-7654 (2019); *ACS Nano* **14**, 13391-13405 (2020)). We thus hypothesize that the actual cause of bacterial death triggered by MPS-PVP is DNA degradation and lipid peroxidation.

To verify this hypothesis, we firstly determined the degree of bacterial genomic DNA degradation after incubation with MPS-PVP in simulated infectious conditions. A notable DNA cleavage effect was exhibited after MPS-PVP treatments (**Figure S3**). Additionally, the MDA content of bacteria exposed to various treatments was quantified. The results show a

statistically significant increase in bacterial MDA level in MPS-PVP groups (as MPS1 and MPS2) (Figure S4), indicating a robust lipid peroxidation. Above supplementary data suggest that the actual route of bacterial death induced by MPS-PVP is DNA degradation and lipid peroxidation.

Figure S3. Genomic DNA degradations of *S. aureus* and *E. coli* with different treatments.

Figure S4. Lipid peroxidations of *S. aureus* and *E. coli* by various treatments. Data are mean \pm s.d. ($n=3$ per group) and n represents biologically independent experiments. Two-tailed Student's t-test was used for statistical analysis. * $P < 0.05$, ** $P < 0.01$.

The antigens released from bacteria are likely to be species non-specific. How would this

affect the normal microbiome in the host?

Response: Thank you very much for the constructive suggestion. We do agree with you about the necessity of investigating potential influence by the antigens released from bacteria upon the normal microbiome in the host. Considering that the gut microbiota features a major constitute of normal microbiome in the host^{4, 5} (*Nat. Rev. Microbiol.* **19**, 55-71 (2021); *Science* **371**, 602-609 (2021)), we therefore have conducted the 16S rRNA-targeted sequencing analysis^{6, 7} (*Nat. Med.* **25**, 448-453 (2019); *Sci. Adv.* **7**, eabf0677 (2021)) to investigate the influence of MPS-PVP on gut microbiota in the revised manuscript (**Figure S16**). Here we compared microbiota isolated from the cecum content of animals between in the control and MPS-PVP group. It could be found that the MPS-PVP treatment exhibits a negligible effect on gut microbiome as can be known from the highly similar microbial compositions between MPS-PVP treatment and PBS groups, which further proves the biosafety of MPS-PVP therapy.

Figure S16. 16S rRNA-targeted sequencing of gut microbiota of mice with *in situ* implant infection model after the MPS-PVP treatment. **a**, Venn diagram of detected bacterial OTUs (operational taxonomic units) in cecum contents of mice after treatment with PBS (Control)

or MPS-PVP, respectively (12 mice per group). **b**, Taxonomic analysis at the phylum level of the gut microbiota from mice with different treatments. **c**, Alpha diversity was presented with ACE to compare the species richness of individual specimens. **d**, Beta diversity was presented with principal coordinates analysis (PCoA) to reflect the species diversity in community composition and structure between different groups. **e**, Analysis of top 5 differentially abundant phylum in different groups.

L44-46. This sentence is not clear and needs to be rephrased.

Response: Thank you very much for your kind reminding. We have rephrased this sentence accordingly in the revised manuscript as follows:

“Biofilm elicits a hypoxic, nutrient-deprived and acidic microenvironment **which polarizes the host immune cells towards anti-inflammatory phenotypes.**” (The relevant changes in bold letters)

L237. What is meant by “... implant biofilm invasive”?

Response: We apologize for the unclear expression and have therefore rewritten this sentence accordingly in our revised manuscript as follows:

“More importantly, **the injected MPS-PVP demonstrates an attractive biofilm eradication benefit in comparison to using vancomycin, thus overcoming the drug-tolerance of implant-related biofilm *in vivo***⁸.” (The relevant changes in bold letters)

References

1. Gao, F., Shao, T., Yu, Y., Xiong, Y. & Yang, L. Surface-bound reactive oxygen species generating nanozymes for selective antibacterial action. *Nat. Commun.* **12**, 745 (2021).

Shanghai Institute of Ceramics, Chinese Academy of Sciences

1295 Dingxi Road, Shanghai 200050, P. R. China.

2. Xi, J. et al. Copper/Carbon Hybrid Nanozyme: Tuning Catalytic Activity by the Copper State for Antibacterial Therapy. *Nano Lett.* **19**, 7645-7654 (2019).
3. Guo, G. et al. Space-Selective Chemodynamic Therapy of CuFe₅O₈ Nanocubes for Implant-Related Infections. *ACS Nano* **14**, 13391-13405 (2020).
4. Fan, Y. & Pedersen, O. Gut microbiota in human metabolic health and disease. *Nat. Rev. Microbiol.* **19**, 55-71 (2021).
5. Baruch, E.N. et al. Fecal microbiota transplant promotes response in immunotherapy-refractory melanoma patients. *Science* **371**, 602-609 (2021).
6. Feehley, T. et al. Healthy infants harbor intestinal bacteria that protect against food allergy. *Nat. Med.* **25**, 448-453 (2019).
7. Lin, S. et al. Mucosal immunity-mediated modulation of the gut microbiome by oral delivery of probiotics into Peyer's patches. *Sci. Adv.* **7**, eabf0677 (2021).
8. Koo, H., Allan, R.N., Howlin, R.P., Stoodley, P. & Hall-Stoodley, L. Targeting microbial biofilms: current and prospective therapeutic strategies. *Nat. Rev. Microbiol.* **15**, 740-755 (2017).

Response to reviewer II.

Comments from reviewer II:

The manuscript by Yang and co-authors describes an inorganic nanosheet of MnPSe₃ as a trigger of anti-biofilm humoral immunity.

Well written, up-to-date introduction, clear results description and high quality images.

Few misspelling words: Giemsa (line 558 and 811); pH and PH (line 416); “the” instead of “he” (line 372)

Response: Thank you very much for the positive comments, suggestions and recommendation. We apologize for the misspelling words and have corrected them in the revised manuscript.

Materials and methods section need some improvement:

Response: Thank you very much for the kind suggestion. We have updated the materials and methods section accordingly in the revised manuscript.

1. please indicate where cells and bacteria cultures were described (line 393)

Response: We have added more detailed description regarding the cells and bacteria culture conditions in the revised manuscript.

2. identify the type and model of centrifuge

Response: The type and model of centrifuge have been supplemented in the revised manuscript.

3. for microscopy studies (confocal and SEM) describe the conditions used

Response: We have added detailed conditions used in microscopy studies in the revised manuscript.

4. line 536, please indicate the suppliers of the mouse cytokine kits

Response: We have provided the suppliers information for the mouse cytokine kits in the revised manuscript.

The inorganic nanosheets (MPS) were combined with a polymer (PVP) to ensure colloidal stability and biocompatibility. So, the cells and tissues are in contact with the MPS-PVP and this should be clear throughout the manuscript, starting with the title. Is there any information regarding the PVP important for the immune response?

Response: Thank you very much for the constructive suggestion. PVP is a typical modification agent for endowing the nanoparticles with good colloidal stability and biocompatibility. According to the reviewer's suggestion, we have clarified the statements about MPS-PVP throughout the manuscript, especially in the phrases about the cells and tissues in contact with MPS-PVP. Besides, we have investigated the specific immune responses triggered by PVP in the revised manuscript (**Figure S6 and S14**) as follows:

Firstly, we investigated whether PVP could affect the immune responses *in vitro* or not. We analyzed the expressions of M1 (CCR7) and M2 (CD206) polarization markers after treating macrophages with varied concentrations of PVPs. As shown in **Figure S6**, the addition of PVPs does not change the phenotype of macrophages. Such a result indicates that MPS, rather than PVP, is critical to provoke the macrophage polarization. In addition, we analyzed the maturation behavior of dendritic cells in the PVPs-treated mice which reflects the possible influences of PVPs on immune responses *in vivo*. As shown in **Figure S14**, administration of PVPs on the infected mice model does not induce percentage fluctuations of mature dendritic cells. These data evidence the critical role of MPS, rather than PVP, in eliciting antibacterial immune responses. Above two sets of data illustrate that PVP plays a

negligible role in provoking immune responses.

Figure S6. Negative immune responses of macrophages by PVPs *in vitro*. Representative flow cytometry plots and quantification analysis showing the phenotype of macrophages (CCR7, M1 phenotype marker. CD206, M2 phenotype marker) after co-culture with different concentrations of PVPs (PVP-1, 10 $\mu\text{g mL}^{-1}$. PVP-2, 50 $\mu\text{g mL}^{-1}$. PVP-3, 100 $\mu\text{g mL}^{-1}$). Data are mean \pm s.d. ($n=3$ per group) and n represents biologically independent experiments. One-way ANOVA with Tukey's post hoc test was used for multiple comparisons. ns, not significant.

Figure S14. Negligible influence of PVPs on dendritic cells (DCs) maturation *in vivo*.

Flow cytometry analysis of mature DCs (CD80⁺CD86⁺ cells in CD11c⁺MHC-II⁺) in IDLN of the mice treated with different concentrations of PVPs (PVP-1, 10 mg kg⁻¹; PVP-2, 20 mg kg⁻¹) in the “*in situ*” implant infection model. Data are mean ± s.d. (*n*=3 per group) and *n* represents biologically independent experiments. One-way ANOVA with Tukey’s post hoc test was used for multiple comparisons. ns, not significant.

RNA-seq reveals MPS-PVP as an essential tool for immune checkpoints in APCs, yet data was obtained only for macrophages. This should be stated in the text, and not only in the materials and methods section.

Response: Thank you very much for pointing out this issue. Accordingly, we have clarified that only macrophages were used as a specific type of APCs for RNA-seq analysis throughout the revised manuscript

REVIEWER COMMENTS

Reviewer #1 (Remarks to the Author):

The authors did a good job in revision. The responses, especially the new experiments, addressed my comments. I have no further concerns.

Reviewer #3 (Remarks to the Author):

The paper entitled: "Inorganic immune checkpoint blockade awakening humoral immunity against medical implant infections" offers an idea on the mechanism of inorganic nanomaterials acting as immune checkpoint blockades (ICB), introducing a readily translatable approach for awakening humoral immunity to eradicate biofilm. This anti-biofilm approach is starting to be much investigated in recent years, as we have entered the post-antibiotic era.

The study is well documented and follows a logical pathway to reach the main hypothesis. Also, authors have improved their work, addressing recommendations of two previous reviewers. Materials and methods section was also updated with some technical details to facilitate understanding.

The bactericidal mechanisms of the MPS via ROS generation have been critically dissected in the revised manuscript, being referred to the related experimental data (which support the hypothesis that DNA degradation is a possible explanation). Aspects regarding the biosafety of MPS-PVP therapy are also described, in terms of microbiota interference, however, additional studies could better clarify their impact not only in microbiota diversity but also against particular phyla, such as those associated with inflammation in the gut, for example. These studies are not the focus of this paper, and I consider that the presented data are sufficient to support the hypothesis formulated by the authors, that MPS-PVP therapy show very low effect against microbiota.

The role of the utilized polymer (PVP) to ensure colloidal stability and biocompatibility is also discussed in the revised version, highlighting that the polarization of macrophages and maturation behavior of dendritic cells are not significantly influenced by PVP. Also, authors have better focused their discussion and presentation of the results to highlight the actual limitations of the study, such as that fact that only macrophages were used as a specific type of APCs for RNA-seq analysis throughout the revised manuscript.

Text was revised and some sentences were rephrased for clarity, misspellings were corrected. However, some spaces and undefined short names (i.e. Abstract: "MPS-PVP" is used without the previous description of the concept). Considering all of the above, I support the acceptance of the paper after another careful read, to ensure that all changes made during revision are correctly incorporated in the final document.

Reviewer #4 (Remarks to the Author):

Yang et al. report that MnPSe3 nanosheets trigger immunity against bacterial biofilm formation. The study is interesting, addresses an important medical problem and suggests novel routes to overcome it. The manuscript is well written. Unfortunately, experimental detail is lacking which makes it difficult to properly interpret the findings.

Specific points:

What kind of macrophages were analyzed by RNAseq in Figure 3? Were they purified ex vivo or generated in vitro or was this the RAW cell line?

Which bacteria were used for infection in figure 5? Was this a combined infection of Staph aureus and E. coli? Did the authors also use other antibiotics or combination of antibiotics as controls?

Vancomycin is neither the antibiotic of choice for treating Staph aureus or E. coli infection, it does not penetrate tissues well and is also not active against bacteria in biofilms or biofilm formation. B cells likely contribute to the anti-bacterial defense. But are antibody responses and titers indeed altered in the absence or presence of nanosheets?

Shanghai Institute of Ceramics, Chinese Academy of Sciences

1295 Dingxi Road, Shanghai 200050, P. R. China.

Response to reviewer #1.

Comments from reviewer #1:

The authors did a good job in revision. The responses, especially the new experiments, addressed my comments. I have no further concerns.

Response: Thank you very much for the positive comment and kind recommendation.

Response to reviewer #3.

Comments from reviewer #3:

The paper entitled: “Inorganic immune checkpoint blockade awakening humoral immunity against medical implant infections” offers an idea on the mechanism of inorganic nanomaterials acting as immune checkpoint blockades (ICB), introducing a readily translatable approach for awakening humoral immunity to eradicate biofilm. This anti-biofilm approach is starting to be much investigated in recent years, as we have entered the post-antibiotic era.

The study is well documented and follows a logical pathway to reach the main hypothesis. Also, authors have improved their work, addressing recommendations of two previous reviewers. Materials and methods section was also updated with some technical details to facilitate understanding.

The bactericidal mechanisms of the MPS via ROS generation have been critically dissected in the revised manuscript, being referred to the related experimental data (which support the hypothesis that DNA degradation is a possible explanation). Aspects regarding the biosafety of MPS-PVP therapy are also described, in terms of microbiota interference, however, additional studies could better clarify their impact not only in microbiota diversity but also against particular phyla, such as those associated with inflammation in the gut, for example. These studies are not the focus of this paper, and I consider that the presented data are sufficient to support the hypothesis formulated by the authors, that MPS-PVP therapy show very low effect against microbiota.

The role of the utilized polymer (PVP) to ensure colloidal stability and biocompatibility is also discussed in the revised version, highlighting that the polarization of macrophages and maturation behavior of dendritic cells are not significantly influenced by PVP. Also, authors have better focused their discussion and presentation of the results to highlight the actual limitations of the study, such as that fact that only macrophages were used as a

specific type of APCs for RNA-seq analysis throughout the revised manuscript.

Text was revised and some sentences were rephrased for clarity, misspellings were corrected. However, some spaces and undefined short names (i.e. Abstract: “MPS-PVP” is used without the previous description of the concept). Considering all of the above, I support the acceptance of the paper after another careful read, to ensure that all changes made during revision are correctly incorporated in the final document.

Response: Thank you very much for the constructive comments and kind recommendations. We apologize for our oversight regarding the undefined short names (i.e. Abstract: “MPS-PVP” is used without the previous description of the concept). We have carefully checked our manuscript and annotated the undefined short names. We have also added discussions regarding the role of APCs including macrophages and dendritic cells in implant infections (**Page 15**). After another careful check, we ensured all changes made during revision are correctly incorporated in the revised manuscript.

Response to reviewer #4.

Comments from reviewer #4:

Yang et al. report that MnPSe3 nanosheets trigger immunity against bacterial biofilm formation. The study is interesting, addresses an important medical problem and suggests novel routes to overcome it. The manuscript is well written. Unfortunately, experimental detail is lacking which makes it difficult to properly interpret the findings.

Response: Thank you very much for the insightful reviews and constructive comments. We apologize for the inadequate experimental details which influenced the proper interpretation of our findings. We have supplemented relevant experimental details according to your suggestions. Please find the following point-by-point detailed responses to your suggestions.

Specific points:

What kind of macrophages were analyzed by RNA-seq in Figure 3? Were they purified ex vivo or generated in vitro or was this the RAW cell line?

Response: Thank you very much for the constructive comments. Raw 264.7 macrophage cell line was used in the RNA-seq analysis in this study. According to your suggestion, we have added detailed descriptions regarding the specific macrophage cell type throughout the revised manuscript.

Which bacteria were used for infection in figure 5? Was this a combined infection of *Staph aureus* and *E. coli*? Did the authors also use other antibiotics or combination of antibiotics as controls? Vancomycin is neither the antibiotic of choice for treating *Staph aureus* or *E. coli* infection, it does not penetrate tissues well and is also not active against bacteria in biofilms or biofilm formation.

Response: Thank you very much for the constructive comments. Methicillin-resistant

Staphylococcus aureus (MRSA, ATCC 43300) was used in the implant infection model as demonstrated in **Figure 5**, and this was not a combined infection. Considering vancomycin is the standard of care for MRSA infection treatment in clinical practice, we used vancomycin as controls to evaluate the antibacterial and anti-biofilm effect of MnPSe₃ nanosheets. The rationale of using MRSA to construct implant infection model *in vivo* and using vancomycin treatment as controls were presented as follows:

Staphylococcus aureus is the main causative bacteria in implant infections, and nearly half of *Staphylococcus aureus* clinical isolates are MRSA in the United States¹⁻³ (*N. Engl. J. Med.* **355**, 666-674 (2006); *Diagn. Microbiol. Infect. Dis.* **93**, 125-130 (2019); *Antimicrob. Agents Chemother.* **65**, e0030321 (2021)). On this ground, we chose a typical MRSA isolate (ATCC 43300) for the construction of implant infection model *in vivo*.

Vancomycin is widely reported as the primary antibiotic in clinic for the management of MRSA infections⁴⁻⁶. Specifically, Argenson *et al.* (*J. Arthroplasty* **34**, S399-S419 (2019)) demonstrated that vancomycin is the primary parenteral agent for MRSA infections recommended by the current Infectious Diseases Society of America guidelines. Hsu *et al.* (*J. Bone Jt. Surg., Am. Vol.* **99**, 223-231 (2017)) have reported that the vancomycin-based bone cement is used in clinic for the treatment of knee periprosthetic joint infection caused by MRSA. Tong *et al.* (*JAMA* **323**, 527-537 (2020)) demonstrated that vancomycin is the current standard therapy for MRSA bacteremia. We agree that vancomycin does not penetrate tissues thoroughly by intravenous administration, therefore it's hard to eradicate biofilm at the infection sites with the intravenous administration-enabled vancomycin dosage. On the contrary, the direct application of vancomycin to the infected area can yield effective biofilm control^{3,7} (Whiteside *et al.* (*Clin. Orthop. Relat. Res.* **469**, 26-33 (2011)) demonstrated that one-stage revision combined with intraarticular vancomycin infusion showed good efficacy in controlling MRSA infections in knee arthroplasty infection patients. Wei *et al.* (*Antimicrob. Agents Chemother.* **65**, e0030321 (2021)) reported that the MRSA periprosthetic joint infection rats treated with intraarticular injection of vancomycin showed better outcomes in infection symptoms, bacterial

counts and biofilm on the prosthesis compared with those treated with systemic vancomycin. Thus, we applied local injection of vancomycin and MnPSe₃-PVP nanosheets at the infection site in our study.

In clinical practice, vancomycin is the widely used classical antibiotic for the treatment of implant infections caused by MRSA^{5, 7-10} (*J. Bone Jt. Surg., Am. Vol.* **99**, 223-231 (2017); *Clin. Orthop. Relat. Res.* **469**, 26-33 (2011); *J. Arthroplasty* **32**, 2508-2512 (2017); *Clin. Orthop. Relat. Res.* **470**, 236-243 (2012); *J. Arthroplasty* **29**, 564-568 (2014)). In the area of basic researches, intra-articular vancomycin powders combined with systemic vancomycin prevented the formation of biofilm on implants in MRSA knee joint infection rat model¹¹ (*J. Bone Jt. Surg., Am. Vol.* **99**, 232-238 (2017)). In another study, one stage revision surgery followed with intra-articular vancomycin powders in combination with subsequent systemic vancomycin that eradicated the as-formed biofilm on implants in the MRSA periprosthetic joint infection rat model³ (*Antimicrob. Agents Chemother.* **65**, e0030321 (2021)). In addition, there are many studies utilizing vancomycin-based therapies for the treatment of MRSA biofilms¹²⁻¹⁴. Suhardi *et al.* (*Nat. Biomed. Eng.* **1**, 1-11 (2017)) reported that vancomycin/rifampin eluting joint implant eradicates *S. aureus* biofilms in rabbit periprosthetic joint infection model. Xi *et al.* (*Nat. Commun.* **12**, 5473 (2021)) demonstrated that vancomycin containing poly(ethylene glycol) and poly(allyl mercaptan) (PEG-PAM) polymers coating successfully prevents implant infection and biofilm formation in both post-arthroplasty infection and post-spinal surgery infection of mice models. Nodzo *et al.* (*Clin. Orthop. Relat. Res.* **474**, 1668-1675 (2016)) demonstrated that cathodic voltage-controlled electrical stimulation combined with the prolonged vancomycin reduced implant bacterial burden of a titanium implant-associated infection in a rat model.

We do agree with you that utilizing vancomycin alone have moderate anti-biofilm effect, and vancomycin is generally used in combination with other antibiotics or surgeries to maximize antibacterial effect. Besides, some patients may experience infection relapse after antibiotic withdraw. Thus, researches developed various alternative therapies to antibiotics,

such as nanoagents and genetic modified immune cells, to treat MRSA infection. Of note, they often use vancomycin as a classical clinical standard of care treatment group, to evaluate the antibacterial and anti-biofilm effect of their proposed alternative therapeutic modes. In one instance, immunomimetic designer cell was introduced to prevent MRSA biofilm formation and cure MRSA implant infection *in vivo*¹⁵ (*Cell* **174**, 259-270 e211 (2018)). Specifically, the conventional vancomycin treatment was used as control group to evaluate the anti-biofilm effect of the immunomimetic designer cell. In another case, zinc-doped Prussian blue nanoparticle was synthesized for the clearance of MRSA infection *via* photothermal effect¹⁶ (*Nat. Commun.* **10**, 4490 (2019)). Vancomycin treatment was also selected as a positive control for the treatment of MRSA-infected wounds. Moreover, researchers evaluated the photo-immunologic therapeutic effect of photosensitizer AgB nanodot for the treatment of MRSA infections by using vancomycin as the positive control¹⁷ (*Adv. Mater.* **34**, e2107300 (2022)). One more example, an enantiomeric block co-beta peptide, poly(amido-D-glucose)-block-poly(beta-L-lysine), was synthesized to eradicate MRSA biofilm¹⁸ (*Nat. Commun.* **10**, 4792 (2019)). In this regard, the authors also used vancomycin as clinical standard of care control group to evaluate the anti-biofilm effect of the co-beta peptide *in vitro* and *in vivo*.

Based on previous studies, we think it's rational to use vancomycin as clinical relevant standard of care group to evaluate the antibacterial and anti-biofilm effect of MnPSe₃ nanosheets. We thank the reviewer for the insightful comments and have added experimental details and discussions in the revised manuscript (**Page 9-10, Page 22, Page 24**, and the figure captions of **Figure 5-6**).

B cells likely contribute to the anti-bacterial defense. But are antibody responses and titers indeed altered in the absence or presence of nanosheets?

Response: Thank you very much for the insightful and constructive comments. In this study, MnPSe₃ nanosheets triggered biofilm destruction and bacterial associated antigens release, then

these endogenous antigens initiated the antibacterial immune responses including B cell activation and antibody production with the help of ICB-mimic MnPSe₃ nanosheets. In our “*in situ*” implant MRSA infection model, MnPSe₃ nanosheets treatment resulted in significant enrichment of plasma cells (CD138⁺CD19⁻) and plasmablasts (CD138⁺CD19⁺) in the infectious microenvironment (**Figure 5j**, **Figure S1** and **Figure S10-11**). Meanwhile, we observed a significant higher levels of serum IgG and IgM in the mice treated with MnPSe₃ nanosheets (**Figure 5n**). Plasma cells are the major effector B cells to produce bacteria-specific neutralizing antibodies in humoral immunity¹⁹ (*Immunity* **45**, 471-482 (2016)). The antibody response is deemed as the key correlate of protection and it's the basis for the neutralization of bacteria in the resistant bacterial infections. However, the development of antibody production is greatly impaired in the presence of biofilm^{20, 21} (*Antimicrob. Agents Chemother.* **60**, 2292-2301 (2016); *Trends Microbiol.* **27**, 303-322 (2019)). Our results suggest that MnPSe₃ nanosheets boost B cell activation and antibody production which is vital for the clearance of biofilm. In our presurgical neoadjuvant immunotherapy model, the treatment with MnPSe₃ nanosheets induced the highest levels of memory B cells (B_{mem}, IgG⁺IgD⁻) infiltrates (**Figure 6i-j** and **Figure S17**). Notably, a significantly higher percentage of CD80⁺ B_{mem}, which is a type of memory B cells differentiating into antibody-forming cells²² (*Nature* **571**, 122-126 (2019)), was observed in MnPSe₃ nanosheets group (**Figure 6k-l** and **Figure S17**). Additionally, a significant higher level of IgG was detected in the mice treated with MnPSe₃ nanosheets (**Figure 6n**). These results further evidence that MnPSe₃ nanosheets elicit long term memory B cell responses.

To profoundly elucidate the mechanism of MnPSe₃ nanosheets in boosting B cell and antibody responses. We discussed the individual role of elemental Mn, Se and P in immunological activities as well the role of nanoparticles containing these elements in altering immune responses as below.

Mn and Se are essential micronutrient required for diverse immunological activities. Mn²⁺ activates anti-viral innate immunity *via* increasing the sensitivity of cyclic GMP-AMP [cGAMP] synthase (cGAS) to double-stranded DNA (dsDNA) and promotes stimulator of IFN genes

(STING) activation²³ (*Immunity* **48**, 675-687 e677 (2018)). Meanwhile, Mn²⁺ also functions as a potent innate immune stimulator by itself, which can directly activate cGAS independent of DNA and trigger a distinct catalytic synthesis of 2'3'-cGAMP, and boost immune response as a novel STING agonist^{24, 25} (*Nucleic Acids Res.* **48**, 4435-4447 (2020); *Cell Rep.* **32**, 108053 (2020)). In another study, it was noted that Mn²⁺ could promote immune responses by facilitating antigen uptake, antigen presentation, germinal center formation, B-cell responses and antibody production *via* the activation of both the cGAS-STING and NLRP3-ASC pathways²⁶ (*Cell. Mol. Immunol.* **18**, 1222-1234 (2021)). More specifically, it was demonstrated that colloidal manganese salt could act as an immune potentiator to stimulate humoral and cellular immune responses, inducing antibody production and CD4⁺/CD8⁺ T-cell proliferation and activation. The colloidal Mn salt shows good adjuvant effects for T cell-dependent antigens, such as ovalbumin, and T cell independent antigens, such as bacterial capsular polysaccharides. Similarly, colloidal Mn salt was used to serve as an adjuvant for rabies vaccines²⁷ (*J. Virol.* **95**, e0141421 (2021)). It turned out that colloidal Mn salt could significantly facilitate the generation of T follicular helper cells, germinal center B cells, plasma cells and rabies virus-specific antibody-secreting cells, consequently improve the immunogenicity of rabies vaccines, and provide better IgG and IgM antibodies protection against virulent rabies virus challenge. Recent advances have proposed that various manganese-based nanoagents could potentiate immunotherapy. In one case, a nanomaterial containing Mn²⁺ and cyclic dinucleotide STING agonists was synthesized²⁸ (*Nat. Nanotechnol.* **16**, 1260-1270 (2021)), demonstrating that Mn²⁺ could augment STING agonist activity to initiate robust anti-tumour immunity. In another study, a nanovaccine comprised of the manganese nanoadjuvant and the Receptor-Binding Domain (RBD) of spike protein of coronavirus was constructed²⁹ (*Nano Today* **38**, 101139 (2021)). Such nanovaccine significantly enhanced RBD-specific IgG and IgM response in mice. Researches also took advantage of the immune-stimulatory effect of Mn²⁺ to induce dendritic cell maturation and elicit pathogen-specific memory B-cell response using manganese dioxide nanoparticles in *S. aureus* infection mice model³⁰ (*Small* **16**, e2000589 (2020)).

Selenium supplementation, for the most part, is immuno-stimulatory, featured by a wide range of parameters including innate immune cell functions, T cell proliferation and B cell activity³¹ (*Antioxid. Redox Signaling* **16**, 705-743 (2012)). In one case, selenium diet increased antigen-specific CD4⁺ T cell responses in mice³² (*J. Nutr.* **140**, 1155-1161 (2010)). In a clinical study, it was demonstrated that selenium supplementation had positive effect on the activation and proliferation of B-cell and antibody responses against the diphtheria vaccine³³ (*Biol. Trace Elem. Res.* **81**, 189-213 (2001)). Another study showed that selenium supplementation promoted glutathione peroxidase 4 (GPX4) expression in T cells, amplified the number of follicular helper T cells and boosted antibody responses in mice and young adults immunized with influenza vaccination³⁴ (*Nat. Immunol.* **22**, 1127-1139 (2021)). Researches also devised selenium-based nanomaterial to potentiate immunotherapy. In recent studies, it was shown that selenium containing nanoparticles could strengthen tumor immunotherapy through the activation of natural killer cells^{35, 36} (*Adv. Mater.* e2108167 (2022); *Adv. Mater.* **32**, e1907568 (2020)). For another example, isoniazid incorporated mannosylated selenium nanoparticles was synthesized for promoting mycobacterium tuberculosis clearance in macrophages³⁷ (*Angew. Chem., Int. Ed. Engl.* **59**, 3226-3234 (2020)). In this study, it was demonstrated that the nanoparticle could not only promote the fusion of mycobacterium tuberculosis into lysosomes in macrophages to initiate the innate immunity for lysosomal destruction of mycobacterium tuberculosis, but also regulate macrophage M1 polarization and cytokine production of host cells to initiate innate immunity for antibacterial inhibition of mycobacterium tuberculosis. In another study, it was shown that oral administration of selenium nanoparticles could induce robust Th1 cytokine pattern and antibody production after Hepatitis B surface antigen vaccination in mouse model³⁸ (*J. Infect. Public Health* **10**, 102-109 (2017)). In a *Vibrio cholerae* whole-cell vaccine mouse model, administration of selenium nanoparticle resulted in robust *Vibrio. cholerae*-specific IgG and IgA antibody responses in serum and saliva³⁹ (*J. Immunol. Res.* **2020**, 8874288 (2020)).

P element also plays important role in immunological activities. In the first example, black

phosphorous nanosheet was synthesized to act as immune-potentiating nanoadjuvant⁴⁰ (*Biomaterials* **273**, 120788 (2021)). It was shown that black phosphorous nanosheet could elicit potent antitumor cellular immunity and significantly augment checkpoint blockade against melanoma in a mouse model. Besides, black phosphorous nanomaterial with protein corona decoration induced calcium influx in macrophages and polarized macrophages for tumor cell eradication^{41, 42} (*Nat. Commun.* **9**, 2480 (2018); *Nanoscale* **12**, 1742-1748 (2020)). In another example, black phosphorus based photothermal therapy triggered the *in situ* release of tumor neoantigens and enhanced antitumor immune responses⁴³ (*Angew. Chem., Int. Ed. Engl.* **59**, 22202-22209 (2020)).

As demonstrated by above previous studies, Mn, Se and P elements and nanomaterials containing these elements showed immune-stimulatory effect in different types of immune cells and animal models, including the enhancement of B cell immunity and antibody responses. To generate humoral and cellular immune responses in mice bacterial infection models, various types of biomaterials have been proposed. For example, a mesoporous silica microparticles vaccine containing pathogen-associated molecular patterns from inactivated bacterial-cell-wall lysates was synthesized⁴⁴ (*Nat. Biomed. Eng.* **6**, 8-18 (2022)). In this work, it was demonstrated that the biomaterial vaccine could elicit bacteria-specific antibody responses and protect mice against bacterial infections. In another example, AgB nanodots was synthesized to induce infection-related immunogenic cell death of bacteria and boost the antibacterial effects of macrophages, dendritic cells, T cells and memory B cells in mice MRSA infection model¹⁷ (*Adv. Mater.* **34**, e2107300 (2022)). For the last example, metal-organic framework encapsulated *E. coli* strain CFT073 significantly enhanced lymph node B cell development and antibody production in a mouse model of bacteremia⁴⁵ (*ACS Nano* **15**, 17426-17438 (2021)).

Therefore, we suppose that the synthetic MnPSe₃ nanosheets containing Mn, Se and P elements could effectively boost the B cell and antibody responses in the implant infection model. We have double checked our experiments and evidenced the consistency in conclusions as demonstrated in the revised manuscript (**Figure 5j, Figure 5l, Figure 5n, Figure 6i-l, Figure**

6n, Figure S10-11 and Figure S17). Besides, we have supplemented the background introduction in our manuscript regarding MnPSe₃ nanosheets triggered B cell activation and antibody responses (**Page 13-14**). We thank the reviewer again for the professional and constructive comments.

References

1. Hischebeth, G.T. et al. Staphylococcus aureus versus Staphylococcus epidermidis in periprosthetic joint infection-Outcome analysis of methicillin-resistant versus methicillin-susceptible strains. *Diagn. Microbiol. Infect. Dis.* **93**, 125-130 (2019).
2. Moran, G.J. et al. Methicillin-resistant S. aureus infections among patients in the emergency department. *N. Engl. J. Med.* **355**, 666-674 (2006).
3. Wei, J., Wen, Y., Tong, K., Wang, H. & Chen, L. Local Application of Vancomycin in One-Stage Revision of Prosthetic Joint Infection Caused by Methicillin-Resistant Staphylococcus aureus. *Antimicrob. Agents Chemother.* **65**, e0030321 (2021).
4. Argenson, J.N. et al. Hip and Knee Section, Treatment, Debridement and Retention of Implant: Proceedings of International Consensus on Orthopedic Infections. *J. Arthroplasty* **34**, S399-S419 (2019).
5. Hsu, Y.H. et al. Vancomycin and Ceftazidime in Bone Cement as a Potentially Effective Treatment for Knee Periprosthetic Joint Infection. *J. Bone Jt. Surg., Am. Vol.* **99**, 223-231 (2017).
6. Tong, S.Y.C. et al. Effect of Vancomycin or Daptomycin With vs Without an Antistaphylococcal beta-Lactam on Mortality, Bacteremia, Relapse, or Treatment Failure in Patients With MRSA Bacteremia: A Randomized Clinical Trial. *JAMA* **323**, 527-537 (2020).
7. Whiteside, L.A., Peppers, M., Nayfeh, T.A. & Roy, M.E. Methicillin-resistant Staphylococcus aureus in TKA treated with revision and direct intra-articular antibiotic infusion. *Clin. Orthop. Relat. Res.* **469**, 26-33 (2011).

8. Nodzo, S.R. et al. The Influence of a Failed Irrigation and Debridement on the Outcomes of a Subsequent 2-Stage Revision Knee Arthroplasty. *J. Arthroplasty* **32**, 2508-2512 (2017).
9. Whiteside, L.A., Nayfeh, T.A., LaZear, R. & Roy, M.E. Reinfected revised TKA resolves with an aggressive protocol and antibiotic infusion. *Clin. Orthop. Relat. Res.* **470**, 236-243 (2012).
10. Roy, M.E., Peppers, M.P., Whiteside, L.A. & Lazear, R.M. Vancomycin concentration in synovial fluid: direct injection into the knee vs. intravenous infusion. *J. Arthroplasty* **29**, 564-568 (2014).
11. Edelstein, A.I. et al. Intra-Articular Vancomycin Powder Eliminates Methicillin-Resistant *S. aureus* in a Rat Model of a Contaminated Intra-Articular Implant. *J. Bone Jt. Surg., Am. Vol.* **99**, 232-238 (2017).
12. Suhardi, V.J. et al. A Fully Functional Drug-Eluting Joint Implant. *Nat. Biomed. Eng.* **1** (2017).
13. Xi, W. et al. Point-of-care antimicrobial coating protects orthopaedic implants from bacterial challenge. *Nat. Commun.* **12**, 5473 (2021).
14. Nodzo, S.R. et al. Cathodic Voltage-controlled Electrical Stimulation Plus Prolonged Vancomycin Reduce Bacterial Burden of a Titanium Implant-associated Infection in a Rodent Model. *Clin. Orthop. Relat. Res.* **474**, 1668-1675 (2016).
15. Liu, Y. et al. Immunomimetic Designer Cells Protect Mice from MRSA Infection. *Cell* **174**, 259-270 e211 (2018).
16. Li, J. et al. Zinc-doped Prussian blue enhances photothermal clearance of *Staphylococcus aureus* and promotes tissue repair in infected wounds. *Nat. Commun.* **10**, 4490 (2019).
17. Tang, H. et al. Photosensitizer Nanodot Eliciting Immunogenicity for Photo-Immunologic Therapy of Postoperative Methicillin-Resistant *Staphylococcus aureus* Infection and Secondary Recurrence. *Adv. Mater.* **34**, e2107300 (2022).
18. Zhang, K. et al. Enantiomeric glycosylated cationic block co-beta-peptides eradicate

- Staphylococcus aureus biofilms and antibiotic-tolerant persisters. *Nat. Commun.* **10**, 4792 (2019).
19. Mesin, L., Ersching, J. & Victora, G.D. Germinal Center B Cell Dynamics. *Immunity* **45**, 471-482 (2016).
 20. Estelles, A. et al. A High-Affinity Native Human Antibody Disrupts Biofilm from Staphylococcus aureus Bacteria and Potentiates Antibiotic Efficacy in a Mouse Implant Infection Model. *Antimicrob. Agents Chemother.* **60**, 2292-2301 (2016).
 21. Raafat, D., Otto, M., Reppschlager, K., Iqbal, J. & Holtfreter, S. Fighting Staphylococcus aureus Biofilms with Monoclonal Antibodies. *Trends Microbiol.* **27**, 303-322 (2019).
 22. Oh, J.E. et al. Migrant memory B cells secrete luminal antibody in the vagina. *Nature* **571**, 122-126 (2019).
 23. Wang, C. et al. Manganese Increases the Sensitivity of the cGAS-STING Pathway for Double-Stranded DNA and Is Required for the Host Defense against DNA Viruses. *Immunity* **48**, 675-687 e677 (2018).
 24. Hooy, R.M., Massaccesi, G., Rousseau, K.E., Chattergoon, M.A. & Sohn, J. Allosteric coupling between Mn²⁺ and dsDNA controls the catalytic efficiency and fidelity of cGAS. *Nucleic Acids Res.* **48**, 4435-4447 (2020).
 25. Zhao, Z. et al. Mn(2+) Directly Activates cGAS and Structural Analysis Suggests Mn(2+) Induces a Noncanonical Catalytic Synthesis of 2'3'-cGAMP. *Cell Rep.* **32**, 108053 (2020).
 26. Zhang, R. et al. Manganese salts function as potent adjuvants. *Cell. Mol. Immunol.* **18**, 1222-1234 (2021).
 27. Wang, Z. et al. Colloidal Manganese Salt Improves the Efficacy of Rabies Vaccines in Mice, Cats, and Dogs. *J. Virol.* **95**, e0141421 (2021).
 28. Sun, X. et al. Amplifying STING activation by cyclic dinucleotide-manganese particles for local and systemic cancer metalloimmunotherapy. *Nat. Nanotechnol.* **16**, 1260-1270 (2021).

29. Wang, Y. et al. Engineering a self-navigated MnARK nanovaccine for inducing potent protective immunity against novel coronavirus. *Nano Today* **38**, 101139 (2021).
30. Wang, C. et al. Photosensitizer-Modified MnO₂ Nanoparticles to Enhance Photodynamic Treatment of Abscesses and Boost Immune Protection for Treated Mice. *Small* **16**, e2000589 (2020).
31. Huang, Z., Rose, A.H. & Hoffmann, P.R. The role of selenium in inflammation and immunity: from molecular mechanisms to therapeutic opportunities. *Antioxid. Redox Signaling* **16**, 705-743 (2012).
32. Hoffmann, F.W. et al. Dietary selenium modulates activation and differentiation of CD4⁺ T cells in mice through a mechanism involving cellular free thiols. *J. Nutr.* **140**, 1155-1161 (2010).
33. Hawkes, W.C., Kelley, D.S. & Taylor, P.C. The Effects of Dietary Selenium on the Immune System in Healthy Men. *Biol. Trace Elem. Res.* **81**, 189-213 (2001).
34. Yao, Y. et al. Selenium-GPX4 axis protects follicular helper T cells from ferroptosis. *Nat. Immunol.* **22**, 1127-1139 (2021).
35. Wei, Z. et al. Selenopeptide Nanomedicine Activates Natural Killer Cells for Enhanced Tumor Chemoimmunotherapy. *Adv. Mater.*, e2108167 (2022).
36. Gao, S. et al. Selenium-Containing Nanoparticles Combine the NK Cells Mediated Immunotherapy with Radiotherapy and Chemotherapy. *Adv. Mater.* **32**, e1907568 (2020).
37. Pi, J. et al. Macrophage-Targeted Isoniazid-Selenium Nanoparticles Promote Antimicrobial Immunity and Synergize Bactericidal Destruction of Tuberculosis Bacilli. *Angew. Chem., Int. Ed. Engl.* **59**, 3226-3234 (2020).
38. Mahdavi, M. et al. Oral administration of synthetic selenium nanoparticles induced robust Th1 cytokine pattern after HBs antigen vaccination in mouse model. *J. Infect. Public Health* **10**, 102-109 (2017).
39. Raahati, Z., Bakhshi, B. & Najari-Peerayeh, S. Selenium Nanoparticles Induce Potent Protective Immune Responses against *Vibrio cholerae* WC Vaccine in a Mouse Model.

- J. Immunol. Res.* **2020**, 8874288 (2020).
40. Li, W.H. et al. Black phosphorous nanosheet: A novel immune-potentiating nanoadjuvant for near-infrared-improved immunotherapy. *Biomaterials* **273**, 120788 (2021).
 41. Mo, J., Xie, Q., Wei, W. & Zhao, J. Revealing the immune perturbation of black phosphorus nanomaterials to macrophages by understanding the protein corona. *Nat. Commun.* **9**, 2480 (2018).
 42. Mo, J., Xu, Y., Wang, X., Wei, W. & Zhao, J. Exploiting the protein corona: coating of black phosphorus nanosheets enables macrophage polarization via calcium influx. *Nanoscale* **12**, 1742-1748 (2020).
 43. Li, Z. et al. Ag(+) -Coupled Black Phosphorus Vesicles with Emerging NIR-II Photoacoustic Imaging Performance for Cancer Immune-Dynamic Therapy and Fast Wound Healing. *Angew. Chem., Int. Ed. Engl.* **59**, 22202-22209 (2020).
 44. Super, M. et al. Biomaterial vaccines capturing pathogen-associated molecular patterns protect against bacterial infections and septic shock. *Nat. Biomed. Eng.* **6**, 8-18 (2022).
 45. Luzuriaga, M.A. et al. Metal-Organic Framework Encapsulated Whole-Cell Vaccines Enhance Humoral Immunity against Bacterial Infection. *ACS Nano* **15**, 17426–17438 (2021).

REVIEWERS' COMMENTS

Reviewer #3 (Remarks to the Author):

Authors have rigorously revised their paper and I have no further concerns regarding this study. I recommend the publication of the paper in current form.

Reviewer #4 (Remarks to the Author):

The authors have addressed nearly all of my concerns and questions. However, they should still discuss the limitations of vancomycin therapy (poor tissue penetration) as detailed in their point by point reply and also mention the local injection of vancomycin in the respective figure legend.

Shanghai Institute of Ceramics, Chinese Academy of Sciences

1295 Dingxi Road, Shanghai 200050, P. R. China.

Response to reviewer #3.

Comments from reviewer #3:

Authors have rigorously revised their paper and I have no further concerns regarding this study. I recommend the publication of the paper in current form.

Response: Thank you very much for the positive comment and kind recommendation.

Response to reviewer #4.

Comments from reviewer #4:

The authors have addressed nearly all of my concerns and questions. However, they should still discuss the limitations of vancomycin therapy (poor tissue penetration) as detailed in their point by point reply and also mention the local injection of vancomycin in the respective figure legend.

Response: Thank you very much for the constructive comments and kind recommendations. We have added detailed discussions regarding the limitations of vancomycin therapy (poor tissue penetration) in the Discussion section (**Page 13**). We have also mentioned the local injection of vancomycin in the respective figure legends (**Figure 6 and Figure 7**).